

# Multi-scale temporal analysis of evaporation on a saline lake in the Atacama Desert

Felipe Lobos-Roco[1,2], Oscar Hartogensis[1], Francisco Suárez[2,3,4], Ariadna Huerta-Viso[1], Imme Benedict[1], Alberto de la Fuente[5], and Jordi Vilà-Guerau de Arellano[1]

[1]Meteorology and Air Quality, Wageningen University, Wageningen, The Netherlands.
[2]Department of Hydraulic and Environmental Engineering, Pontificia Universidad Católica de Chile, Santiago Chile.
[3]Centro de Desarrollo Urbano Sustentable (CEDEUS), Santiago Chile.
[4]Centro de Excelencia en Geotermia de los Andes (CEGA), Santiago Chile.
[5]Department of Civil Engineering, Universidad de Chile, Santiago, Chile.

**Correspondence:** Felipe Lobos-Roco (felipe.lobosroco@wur.nl; felipe.lobos.roco@gmail.com)

**Abstract.**

We investigate how evaporation changes depending on the scales in the Altiplano region of the Atacama Desert. More specifically, the temporal evolution from the climatological to the sub-diurnal scales on a high-altitude saline lake ecosystem. We analyse the evaporation trends over 70 years (1950-2020) at a high-spatial resolution. The method is based on the downscaling of 30-km hourly resolution ERA5 reanalysis data to 0.1-km spatial resolution data, using artificial neural networks to analyze the main drivers of evaporation. To this end, we use the Penman open water evaporation equation, modified to compensate for the energy balance non-closure and the ice cover formation on the lake during the night. Our estimation of the hourly climatology of evaporation shows a consistent agreement with eddy-covariance measurements and reveals that evaporation is controlled by different drivers depending on the time scale. At the sub-diurnal scale, mechanical turbulence is the primary driver of evaporation, and at this scale, it is not radiation-limited. At the seasonal scale, more than 70% of the evaporation variability is explained by the radiative contribution term. At the same scale, and using a large-scale moisture tracking model, we identify the main sources of moisture to the Chilean Altiplano. In all cases, our regime of precipitation is controlled by large-scale weather patterns closely linked to climatological fluctuations. Moreover, seasonal evaporation influences significantly the saline lake surface spatial changes. From an interannual scale perspective, evaporation increased by 2.1 mm per year during the entire study period, according to global temperature increases. Finally, we find that yearly evaporation depends on the El Niño Southern Oscillation (ENSO), where warm and cool ENSO phases are associated with higher evaporation and precipitation rates, respectively. Our results show that warm ENSO phases increase evaporation rates by 15%, whereas cold phases decrease it by 2%.

## 1 Introduction

In arid regions, evaporation is one of the most important components in the water cycle since potential evaporation is typically one order of magnitude larger than precipitation (Lictevout et al., 2013; Houston, 2006). Investigating evaporation in





these regions is challenging due to the lack of observations, the landscape complexity, and the poor representation in hydrom-eteorological models. The climate/large-scale atmospheric circulation and spatially localized zones affect water availability (Lobos-Roco et al., 2021). At a local level in the Atacama Desert, evaporation occurs (Houston, 2006): i) in rivers and the adjacent riparian zones; ii) in marshlands, where localized groundwater springs support vegetation growth and sometimes con-tribute to the formation of shallow terminal lakes (de la Fuente and Meruane, 2017), which generally occurs in the Andes Mountains; iii) in salt flats or playas, which are the result of more extensive groundwater discharge in endorheic basins (de la Fuente, 2014; de la Fuente and Meruane, 2017; Suárez et al., 2020); and iv) in bare soils where the water table is shallow (Rosen, 1994; Johnson et al., 2010; Uribe et al., 2015; Blin et al., 2022). The Chilean Altiplano is an arid zone where water evaporates from spatially localized environments, removing water from the basin. The Altiplano region has a unique environ-mental, economic, and social value due to its location within the Atacama Desert where rainfall provides a source of water for northern Chile. A reliable understanding of the processes that govern evaporation in this region is essential for three main reasons (Suárez et al., 2020). First, water resource management because a correct quantification of these fluxes enhances the performance of water balance models and improves the estimation of the basin's water recharge. Second, terrestrial and aquatic ecosystems that sustain the native flora and fauna of this region. Third, sustainable agricultural and mining production in terms of minimizing environmental impacts and maximizing water use. Within the Altiplano, the Salar del Huasco basin is chosen for studying evaporation due to the perennial terminal saline lake where non-local atmospheric processes occur (Suárez et al., 2020; Lobos-Roco et al., 2021). This lake, untouched by human activities (Uribe et al., 2015), has been well-studied in recent years. These studies have focused on quantifying and understanding evaporation (de la Fuente and Meruane, 2017; Suárez et al., 2020; Lobos-Roco et al., 2021, 2022) for use in water resource management models (Uribe et al., 2015; Blin et al., 2022). Thus, there are many comprehensive datasets of surface and upper-atmospheric observations for the Salar del Huasco basin that can be used to relate large-scale atmospheric phenomena with small-scale processes and, in turn, to advance our understanding of evaporation and its use for water resource conservation.

Synoptic and regional circulation over the Altiplano region, responsible for moisture transport and precipitation, has been studied by Rutllant et al. (2003), Falvey and Garreaud (2005), and Böhm et al. (2020). These studies investigated how large-scale atmospheric phenomena influenced by the Pacific Ocean, steep Andean topography, and the Amazon basin organize circulations at different scales. These atmospheric circulations are the main contributors of moisture in the region. Two marked phases characterize the principal synoptic atmospheric circulation over the Altiplano region. The first phase occurs during the summer season (December to March). It is characterized by westward winds from the Amazon basin, which transport a significant amount of moisture over the Altiplano (Falvey and Garreaud, 2005). This moisture transport is highly variable from year to year, and it is responsible for convective rains that occur in the region. In the second phase dry air from the free troposphere above the Pacific Ocean is transported to the Altiplano region in the Andes (Rutllant et al., 2003) and occurs from April to November. This dry air transport results from the thermal differences between the western slope of the Atacama Desert and the Pacific Ocean (Lobos-Roco et al., 2021). Other studies have reported the effects of the El Niño Southern Oscillation (ENSO) and the Pacific Decadal Oscillation (PDO) on precipitation patterns. Böhm et al. (2020) studied the integrated water vapor (IWV) variability and its relationship with the ENSO phenomenon in the Atacama Desert during the $20^{th}$ century. Their




results revealed that cool ENSO phases (associated with the La Niña ENSO phenomenon) yield greater IWV variability which favors more extreme wetter conditions during the austral summer in the Altiplano region. Garreaud et al. (2003) analysed the climatic conditions from interseasonal to glacial-interglacial timescales. Researchers found that mean zonal airflow over

the region modulates interannual changes in the climatic condition over the Altiplano. This airflow respond to sea-surface temperature variability in the tropical section of the Pacific Ocean. Likewise, several studies have pointed out the remarkable control that cool ENSO phases exert over the precipitation in the Altiplano (Aceituno, 1988; Vuille et al., 2000; Garreaud and Aceituno, 2001). This control shows that cool ENSO phases yield wetter rainy seasons, whereas warm ENSO phases (El Niño) results in drier rainy seasons (Garreaud and Aceituno, 2001). This dependence on climatic factors means that temperature-

dependent evaporation occurring at local scales is related to such phenomenon.

The spatiotemporal evolution of evaporation has also been investigated in the Altiplano region of the Atacama Desert. These studies aimed to understand the complex diurnal land-atmosphere turbulent transport over different surfaces (Kampf et al., 2005; de la Fuente and Meruane, 2017; Lobos-Roco et al., 2021), characterizing the larger scale influence on the local evaporation (Suárez et al., 2020; Lobos-Roco et al., 2021) or simply to assess daily evaporation from bare soils in order to develop

relationships that can be used to relate evaporation with the water table depth (Johnson et al., 2010). These investigations mainly focused on short-term field experiments based on either daily measurements (Kampf et al., 2005; Suárez et al., 2020) or applied models used to predict potential evaporation (de la Fuente and Meruane, 2017). Even with these studies, long-term evaporation observations at a local scale are still lacking, especially when trying to construct conceptual models that can be used for water resource management. For instance, Uribe et al. (2015) developed a hydrological model in the Salar del Huasco region where

evaporation was estimated using information from evaporation pans, and regional vertical gradients in evaporation were seen as a function of elevation. Blin et al. (2022) developed a groundwater model for the Salar del Huasco. This groundwater model, which was used to assess climate change impacts on the Salar del Huasco basin, utilized the hydrological model constructed by Uribe et al. (2015) to determine aquifer recharge and to estimate the evaporation discharge to the atmosphere. Unfortunately, these models overlook the influence of the non-local atmospheric processes, such as the entrainment and advection of heat,

moisture and momentum, on evaporation rates (Suárez et al., 2020; Lobos-Roco et al., 2021, 2022). Attempting to rectify this oversight, recent experimental field campaigns have been carried out in the Altiplano area of the Atacama Desert (Suárez et al., 2020). However, the lack of reliable long-term actual evaporation estimates still limits our complete understanding of the climate change impacts on water availability in these arid areas. Moreover, there are no studies that aim to investigate the myriad of links between these temporal short and large-scale studies. Thus, our objective is to understand seasonal and interannual

evaporation variability by examining how surface energy partitioning, turbulence, and moisture supply affect seasonal changes in evaporation. In this way, we aim to bridge this cross-scale knowledge gap which will help to address water availability in the Atacama Desert.

In this study, we applied climatologically robust downscaled reanalysis data to the saline lake of the Salar del Huasco. Although we focused on one particular saline lake, this kind of surface represents the main evaporation pathway of the Altiplano

region (Houston, 2006). We hypothesized that radiative and aerodynamic components could represent the evaporation of the specific conditions of the saline lake. This representation is performed using an adapted version of the Penman (1948) equation.



The confirmation of this hypothesis enables to extend our evaporation calculations to the entire climatological period (1950-2020) and to investigate evaporation fluctuations in different ENSO phases. To complete this analysis, we show how monthly and yearly evaporation and precipitation lead to changes around the saline lake in the Salar del Huasco. Since precipitation in

this area is closely related to large-scale atmospheric sources, we tracked the origins of these sources for the Salar del Huasco. In our analysis, we applied the following methodological steps. First, we downscaled the reanalysis meteorological data (∼30 km) to local conditions (∼100 m) observed above the saline lake by applying artificial neuronal networks and validating the downscaled data with eddy-covariance (EC) observations. Second, we developed a site-adapted version of the Penman (1948) equation. Third, we analysed under a seasonal perspective the changes in evaporation and its drivers, the sources of

moisture transported into the region, and impacts of evaporation in the saline lake's water balance. Finally, we quantified the climatological trends and interannual evaporation and precipitation anomalies related to ENSO phases and PDO over the last 70 years.

## 2 Methods

### 2.1 Study area

Our study area is located in the Salar del Huasco (SDH) basin (1,462 km$^2$), whose highest point sits 5,200 m above sea level (asl), and its lowest point at 3,790 m asl (Uribe et al., 2015). This endorheic basin is located to the west of the Andes, 135 km inland from the Pacific Ocean. Since the basin is so close to the ocean, an intense and recurrent afternoon atmospheric flow from the ocean transports relatively cold and humid air into the Altiplano (Lobos-Roco et al., 2021). The basin is also affected by the moist atmospheric flow coming from the east which is responsible for a marked rainy season during austral summer,

where short convective storms are the main source of aquifer recharge (Blin et al., 2022). Since evaporation occurs where there is available water, our research focused on the basin's sink, which is a wetland in the Salar del Huasco (de la Fuente et al., 2021). Specifically, our attention is placed on the saline lake of the Salar del Huasco (20.2 °S, 68.8 °W, 3790 m asl), which is a perennial water body surrounded by salt crusts, zones with native vegetation patches and zones with bare soils (Fig. 1). This terminal lake shows significant seasonal changes in its surface ranging between ∼0.5 to 5 km$^2$, and has a measured depth

of ∼15 cm (Lobos-Roco et al., 2021). These types of groundwater-fed wetlands are commonly found in the Altiplano region (Kampf et al., 2005), and result in unique ecological habitats for endemic flora and fauna (Dorador et al., 2013).

### 2.2 Data acquisition

This study combines data from different sources including observations, modelling reanalysis data, and remote sensing datasets. Table 1 summarizes the datasets, variables, frequency, spatial resolution and sources employed in this research.

120       Two in-situ observation datasets are used. The first dataset corresponds to measurements integrated at 10-min intervals, installed ∼1 m above the saline lake of the Salar del Huasco (20.27 °S, 68.88 °W; 3790 m asl) between November 13$^{th}$ and 24$^{th}$, 2018 during the E-DATA field experiment (Suárez et al., 2020). Latent heat ($L_vE$) data were collected from an EC



**Figure 1.** (a) Salar del Huasco saline lake and location of the meteorological station and the eddy covariance (EC) system used in this investigation. The red square shows an approximated grid size of the ERA5 reanalysis data. (b) Schematic cross-section of the meteorological downscaling from larger to smaller spatial scales. Contains modified Copernicus Sentinel data processed by Sentinel Hub, ESA.





**Table 1.** Description of the data used in this research. Variables analysed are incoming shortwave radiation ($Sw_{in}$), net radiation ($R_n$), latent heat flux ($L_vE$), air temperature ($T$), air pressure ($P$), relative humidity ($RH$), specific humidity ($q$), wind speed ($U$), wind direction ($WD$), zonal wind ($u$), meridional wind ($v$), Pacific Decadal Oscillation (PDO), and Oceanic El Niño Index (ONI).

| Type | Period | Variables | Height | Time frequency | Spatial resolution | Source |
|---|---|---|---|---|---|---|
| EC observations | 13/11/2018 24/11/2021 | $R_n, L_vE, T,$ $P, RH, U, WD$ | 1 m | 10-min | ~100m | E-DATA field experiment (Suárez et al. (2020); Lobos-Roco et al. (2020) |
| Met. st. observations | 1/1/2016 31/12/2019 | $Sw_{in}, T, P,$ $RH, U, WD$ | 2 m | 1 hour | ~100m | Salar del Huasco Meteorological st. from CEAZA (met-station$_{SDH}$) |
| Reanalysis | 1/1/1950 31/12/2020 | $Sw_{in}, T, P,$ $q, U, WD, Pp$ | 2/10 m | 1 hour | ~30 km | Hersbach et al. (2020) |
| Modeling (WAM-2layers) | 1/1/1997 31/12/2018 | $E, Pp, q,$ $u, v$ | 1000-100 hPa | 6-3 hours | 1.5 degrees | Dee et al. (2011) |
| Remote sensing | 1/1/1985 31/12/2020 | SDH lake's area | - | 1 month | 30 m | de la Fuente et al. (2021) |
| Other | 1/1/1950 31/12/2020 | PDO, ONI | - | 1 month | 50°N-50°S 120°-170° W | NCEP-NOAA |

system (EC$_{water}$ in Fig. 1) and meteorological variables, such as net radiation ($R_n$), air temperature ($T$), atmospheric pressure ($P$), relative humidity ($RH$), and wind speed ($U$) and direction ($WD$) were measured using an accompanying weather station
to the EC system (Suárez et al., 2020). The second dataset corresponds to 1-hour measurements collected at the Salar del Huasco meteorological station (met-station$_{SDH}$, Fig. 1; Table 1), which belongs to the Center for Advanced Studies of Arid Zones (CEAZA). This station has been in continuous operation since October 2015, but we use the data from January 2016 to December 2019 (Table 1). The meteorological station is located 2 km North (20.25 °S, 68.87 °W 3,800 m asl) of the EC system, over bare soil at a height of 2 m (Fig. 1b). This dataset ensures an adequate characterization of the diurnal variability
for a relatively long period of 4 years.

The long-term climatological ERA5 reanalysis dataset (Table 1; Hersbach et al. (2020)), available at 1-hour resolution and 30 km spatial resolution, is downscaled to the conditions observed at met-station$_{SDH}$ (section 2.3.1). We use the data corresponding to the grid point of Salar del Huasco at the first level (2-10 m above the surface) from 1950 to 2020. The ERA5 dataset combines a vast amount of historical surface and satellite observations into global estimates with the help of advanced
atmospheric modeling and data assimilation systems (Hersbach et al., 2020). Additionally, we use ERA-interim data (Dee et al., 2011) from 1997 to 2018, at 1.5 degrees of spatial resolution to track the moisture sources (Section 2.3.4). resulting





in precipitation over the region. This data is obtained at 6-hourly timestep for the atmospheric variables (wind and specific humidity) and 3-hourly timestep for the surface variables (evaporation and precipitation).

To obtain the temporal evolution of the water surface of the Salar del Huasco lake, we use the data provided by de la Fuente
et al. (2021). In brief, the saline lake water surface is calculated using Landsat 5 (January 1985 - June 2013) and Landsat 8 (March 2015 - December 2019) satellite images through normalized differenced water index (NDWI) at a pixel resolution of $30 \times 30$ m. The NDWI threshold is adjusted manually and contrasted to the size of the wetland computation based on the NDWI.

Two climatological oceanic indices at a monthly resolution are used to analyze macroclimatic phenomena, such as ENSO
and PDO. These indices are obtained from the National Climate Prediction Center (NCEP). The first one is the Oceanic El Niño Index (ONI), which corresponds to sea surface temperature anomalies in the El Niño 3.4 region (50 °N-50 °S, 120°-170° W) from 1950 to 2020. The second one corresponds to the HC300-based PDO index, a temperature anomaly index based on the heat content anomalies in the first 300 m layer depth of the North Pacific region, 20°N poleward (Kumar and Wen, 2016).

### 2.3   Data processing

#### 2.3.1   Downscaling of meteorological data

The long-term ERA5 data are downscaled from $\sim$30 km to the local conditions ($\sim$10-100 m) observed at the CEAZA's meteorological station (met-station$_{SDH}$ in Fig. 1a). Downscaling is performed using artificial neuronal network (ANN) algorithms (Dibike and Coulibaly, 2006; Kumar et al., 2012). The ANNs are solved using 10-hidden layers and using the Levenberg-Marquardt training algorithm. This process is performed with MATLAB's Neural Fitting tool. Air temperature ($T$), specific
humidity ($q$), and wind speed ($U$) from the ERA5 dataset are used as input data for training and validation of the ANNs, whereas $T$, $RH$, $U$, $WD$, and $Sw_{in}$ collected at met-station$_{SDH}$ are used as target data. Note that conditions observed at the met-station$_{SDH}$ (2 m) shows the same variabilities and magnitudes as the meteorological observations obtained by the EC$_{water}$ above the saline lake (1 m, see Fig. 1b) during the E-DATA field experiment.

As validation, Figures 2 and 3 show the time evolution and orthogonal regression of the ERA5 downscaled and raw variables
of $T$, $q$, and $U$ compared to surface observations of the met-station$_{SDH}$ and EC$_{water}$. In terms of temperature, Figure 2a shows that there are significant differences in the diurnal cycle of $T$ between the ERA5$_{raw}$ data and the observations of met-station$_{SDH}$ and EC$_{water}$, especially at lower temperatures. Nonetheless, the temperatures observed above the water are in agreement with the values from EC$_{water}$ (1 m) and met-station$_{SDH}$ (2 m). Therefore, we can assume that air temperatures above the water and above the land are similar. This similarity allows us to validate ERA5$_{down}$ results on the saline lake
using the data observed by the met-station$_{SDH}$. Figure 3a shows a satisfactory correlation between the ERA$_{raw}$ and the met-station$_{SDH}$ observations ($R^2$ = 0.95) but a low slope (m = 0.5). This mismatch is overcome when we apply the downscaling, where $T$ increases the correlation coefficient ($R^2$ = 0.97) and the slope (m = 0.92) (Fig. 3a).

For specific humidity, there is more scatter in the met-station$_{SDH}$ observations, which results in low $R^2$ = 0.38 (Fig. 3b). However, similar to temperature, we observe an improvement after the downscaling, where ERA5 data increases the slope in





the orthogonal regression from 0.43 to 0.77. Although the agreement between $q$-ERA5 and observations is lower than $T$-ERA5, $q$-ERA5 has a reasonable agreement with observations in the diurnal cycle (Fig. 2b).

**Figure 2.** (a) Data comparison of temperature, (b) specific humidity, and (c) wind speed between the available data sources: observations gathered from an Eddy Covariance (EC) over water surface ($EC_{water}$); observations collected from a meteorological station overland (met-station$_{SDH}$); ERA5 reanalysis raw data (ERA5$_{raw}$); and ERA-5 reanalysis downscaled data (ERA5$_{down}$).

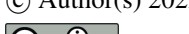



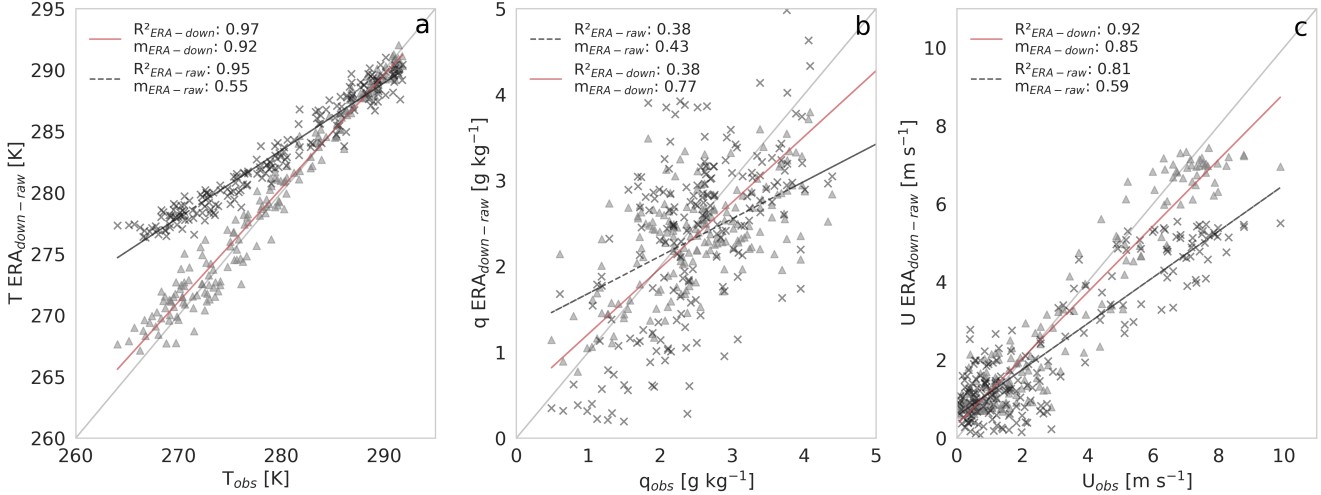

**Figure 3.** Comparison between ERA-5 data before and after the downscaling against the met-station$_{SDH}$ observations for (a) temperature ($T$), (b) specific humidity ($q$), (c) and wind speed ($U$). Crosses represent the ERA5$_{down}$ data and triangles the ERA5$_{raw}$ data.

In terms of wind speed ($U$), we observe more differences between EC$_{water}$ and met-station$_{SDH}$ during the maximum values. These differences are related to the nature of the surface where both instruments take measurements, EC$_{water}$, and met-station$_{SDH}$ above bare soil (Fig. 1). However, our statistical calculations corroborate the benefits of using the downscaling methods: R$^2$ increases from 0.81 to 0.92 and slopes from 0.59 to 0.85 as compared to ERA$_{raw}$. Finally, although wind direction is not used to estimate the evaporation and not shown in the plots, the ERA5 data have good agreement with observations.

### 2.3.2 Actual evaporation estimation

To estimate the actual evaporation, we employ an adapted version of the Penman (1948) equation for open water evaporation (Huerta-Viso, 2021), expressed in energy terms $L_v E$. Our approach is to use standard meteorological data of $T$, $q$, $U$, and $Sw_{in}$ from the downscaled ERA5 dataset, and apply it to the specific conditions of the Salar del Huasco shallow lake. The adapted version of the Penman equation reads as:

$$L_v E = c_{ice} \frac{s}{s+\gamma} c_{EBNC} \overbrace{(R_n - G)}^{\text{Radiative}} + \frac{\rho_a c_p}{s+\gamma} \overbrace{\frac{1}{r_a}(e_s - e)}^{\text{Aerodynamic}}, \tag{1}$$

where $s$ [Pa K$^{-1}$] is the slope of saturated vapor pressure curve, $\gamma$ [Pa K$^{-1}$] is the psychrometric constant, $R_n$ [W m$^{-2}$] is the net radiation, $G$ [W m$^{-2}$] is the ground heat flux, $\rho$ [kg m$^{-3}$] is the dry air density, $c_p$ [J K$^{-1}$ kg$^{-1}$] is the air's specific heat at constant pressure, $r_a$ [s m$^{-1}$] is the aerodynamic resistance, $e_{sat}$ [Pa] is the saturated vapor pressure, and $e_a$ [Pa] is the vapor pressure at measured level. $c_{ice}$ [-] is the ice coefficient, which is a correction coefficient that represents the evaporation reduction that occurs when an ice cover is formed above the saline lake (Vergara-Alvarado, 2017), and $c_{EBNC}$ [-] is the energy



balance non-closure coefficient, which corrects the available energy ($R_n$–$G$) to improve the energy balance closure. Note that Equation 1 becomes the Penman (1948) equation when $c_{ice} = c_{EBNC} = 1$. Appendix A describes the details of the calculation for each term in the Equation 1.

### 2.3.3 Climatological analysis

To evaluate the diurnal variability of evaporation, the evaporation estimates are compared with observations using orthogonal regression, where we estimate the error employing the root means squared error (RMSE), the mean absolute error (MAE), and the correlation (R) and determination ($R^2$) coefficients. The climatology of evaporation estimates and precipitation data obtained from ERA5 is analysed at seasonal and interannual scales. For seasonal time scales, we use descriptive statistics of mean, maximum, minimums, and quantile 25, 50, and 75 for each averaged month over the entire period (1950-2020). For the interannual time scales, we calculate monthly anomalies as the difference between the 12-month moving average and the mean of the entire period under study. Our reason for using the moving average is to decrease the high scatter that monthly means produce and better evaluate the ENSO and PDO influence on the evaporation and precipitation.

### 2.3.4 Large-scale moisture transport tracking model

To get an overview of the moisture transport that result in precipitation over the Altiplano region and surrounding areas, we determine the moisture sources of a selected region. This selected region encompasses the Salar del Huasco and an extensive region around it, spanning from 83° W to 57° E, and from 11° N to 27° S (Fig. 7). Precipitation over this region is tracked backwards in time to determine where the water originally evaporated, the moisture sources. To determine these moisture sources, we use ERA-Interim data (Dee et al., 2011) from 1997-2018 to force the Water Accounting Model-2layers (WAM-2layers; van der Ent et al. (2010); van der Ent (2014)). WAM-2layers is an Eulerian offline moisture tracking model which solves the atmospheric water balance for every grid cell. Tracking is performed on two layers in the atmosphere, hence the atmospheric input variables from ERA-Interim are integrated over two layers. Well-mixed conditions are assumed for both layers. More information on the model is given by van der Ent et al. (2010); van der Ent (2014). Seasonal averages of moisture sources are shown (1997-2018; summer (JFM), autumn (AMJ), winter (JAS), and spring (OND)) together with the direction and intensity of the moisture fluxes.

### 2.3.5 Estimation of the long-term mass balance of the lake

The long-term water balance in the saline lake is assessed by combining the mass conservation principle with actual evaporation estimates and precipitation data. Evaporation estimates are obtained from the downscaled ERA5 and precipitation data from the raw ERA5, both at the grid point of Salar del Huasco (Fig. 1a). The mass balance is evaluated as follows. First, the volume of the lake in a specific month is estimated using the lake's area and assuming a constant lake depth that varied between 0.05 and 0.20 m. Second, we estimate the monthly lake outflow using the actual evaporation values and the lake's area, assuming no groundwater outflow (endorheic basin). Third, we determine the volume reduction of the lake due to evaporation by subtracting





the volume of water evaporated in a month from the volume of the lake. Then, the lake area of the next month is computed

dividing the lake's volume by its depth. This area is compared to that obtained using remote sensing data to determine the additional monthly water volume required to achieve the observed lake surface. By associating this additional water input with precipitation, we determine the areal extension of precipitation that contributes to represent the observed areas of the lake. Because most of the time there are no surface water inputs, this additional water source must represent groundwater inputs into the lake. The approach followed here is a first order approximation that can be used to understand the key components of the

lake's water balance.

## 3 Results and discussion

This section describes the diurnal, seasonal and interannual variability of evaporation at the saline lake of Salar del Huasco. First, we analyse the site-adapted Penman equation evaporation estimates using the $L_v E$ observations taken above the saline lake with an EC as reference. Secondly, we analyze the seasonal variations of evaporation and its main drivers. In addition, we

quantify the role of evaporation in the water balance of the saline lake, including precipitation as an essential component of the water cycle. Finally, we close the article by studying the climatological trends of evaporation-precipitation and the influence of macroclimatic effects such as the ENSO and PDO phenomena on their anomalies.

### 3.1 Diurnal cycle perspectives of evaporation

Within this subsection, we quantify the diurnal cycle of actual evaporation from its energy and aerodynamic contribution using

the standard Penman (1948) equation. In addition, we validate the ice coefficient ($c_{ice}$) and the energy balance non-closure coefficient ($c_{EBNC}$) compared to the evaporation measurements from E-DATA experiment.

Figure 4 shows the averaged $L_v E$ diurnal cycle over the E-DATA period observed by the $EC_{water}$ calculated using the site-adapted Penman equation ($P_{SDH}$, Eq. 1), and the standard Penman (1948) equation ($P_{stdr}$). Figures 4a and 4b indicate that there is a satisfactory agreement between $L_v E$ observed and estimated. The main difference is the two-hour lag during the

morning transition (between 11:00 and 13:00 LT) that results from the height at which ERA5 wind is calculated: 10 m. These data have a root mean square error (RMSE) of 73 W m$^{-2}$, and a mean absolute error (MAE) of 17 W m$^{-2}$. Likewise, the orthogonal regression of $L_v E$ between the $P_{SDH}$ and $EC_{water}$ observations have acceptable correlation and determination coefficients (R = 0.88 and R$^2$ = 0.78, respectively) and orthogonal regression slopes (m = 0.98).

To better understand the $L_v E$ results obtained by $P_{SDH}$, we analyze the radiative energy and aerodynamic contributions

to the standard Penman $L_v E$ separately, along with the performance of the introduced and coefficients. Figure 4c shows the averaged diurnal cycle of the energy and aerodynamic term of the standard Penman equation, compared to the results of $P_{SDH}$ (Eq. 1) and the EC observations of $L_v E$. The $L_v E$-diurnal pattern shows two distinct regimes: in the morning (before 12:00 LT), the aerodynamic term follows the observations closely whereas in the afternoon (after 12:00 LT), the energy term is the one with a closer match. Our explanation is based on the limiting regimes, which have been studied by Lobos-Roco

et al. (2021), Lobos-Roco et al. (2022), and Suárez et al. (2020). During the morning, $L_v E$ is limited by the absence of





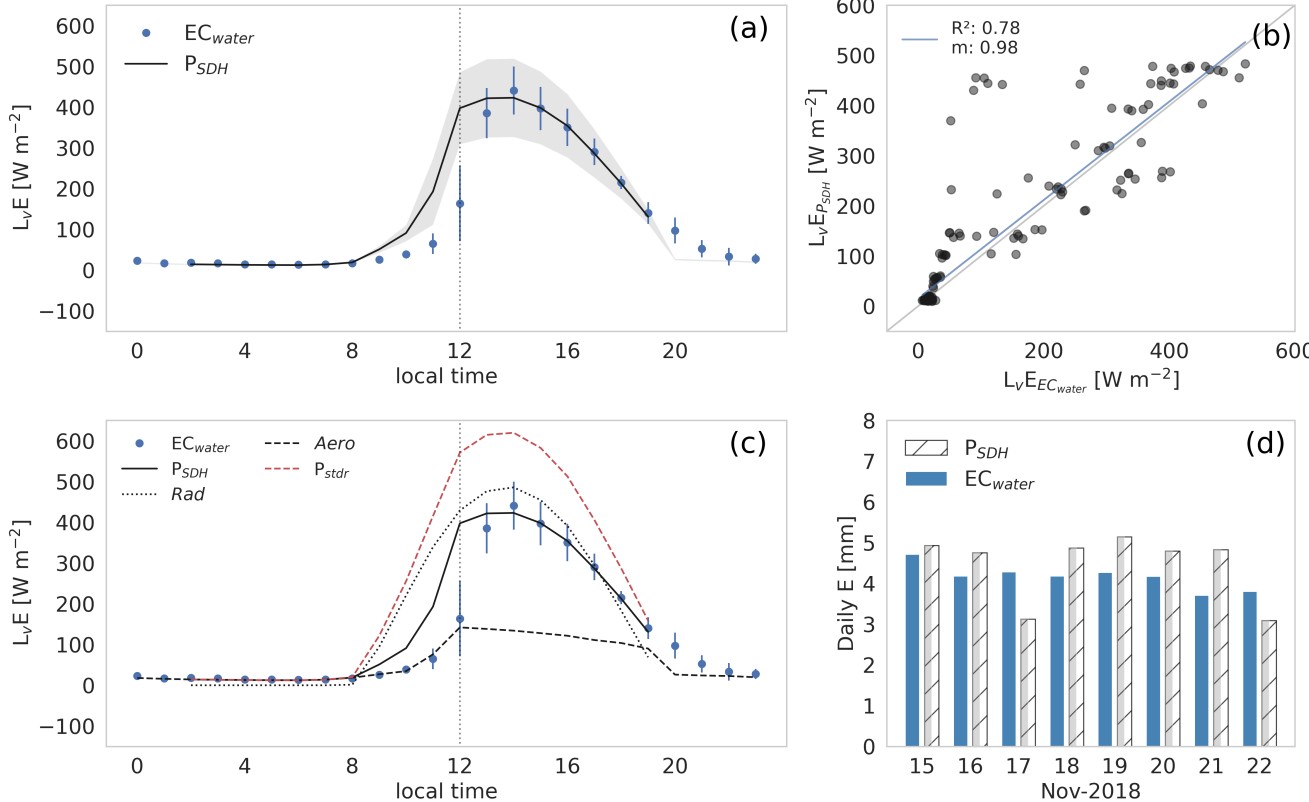

**Figure 4.** (a) E-DATA period averaged and standard deviation of the diurnal cycle of $L_vE$ observed by the $EC_{water}$, calculated by $P_{SDH}$ equation. (b) Orthogonal regression between $L_vE$ measured by the $EC_{water}$ and those estimated through $P_{SDH}$. (c) Diurnal cycle of $L_vE$ observed by the $EC_{water}$, calculated by the $P_{SDH}$, standard Penman ($P_{stdr}$), and the aerodynamic ($Aero$) and radiative ($Rad$) contribution. (d) Daily evaporation (mm) measured by the EC system and estimated through $P_{SDH}$. The vertical dotted line in (a) and (c) indicates the wind regime change.

mechanical turbulence. As a result, the transport from the saturated air above the surface into the dry atmosphere is hampered which results in relatively low values for $L_vE$. In turn, and during the afternoon, due to the regional wind flow arrival, the enhancement of mechanical turbulence leads to high values of evaporation, and $L_vE$ depends on the amount of the available energy. This radiative energy control is more clearly observed from 14:00-15:00 LT when the radiation decreases yields of

$L_vE$. The addition of energy and aerodynamic contribution to the standard Penman equation shown in Figure 4c (dashed red line) demonstrates an overestimation of 88 W m$^{-2}$ concerning the observations, where the diurnal cycle is only followed during the afternoon (windy regime). When comparing the $P_{SDH}$ (Eq. 1) and the standard Penman $L_vE$ equation, we observe that and coefficients significantly improve the evaporation estimates. This improvement is given first by the coefficient that reduces the available radiative energy under calm wind conditions, decreasing it by 70% and 30% under windy conditions.





**Table 2.** Statistical metrics for comparing standard Penman, site-adapted Penman, radiation and aerodynamic contribution for $L_vE$, compared to $L_vE\ EC_{water}$ observations. Monthly evaporation integration compares site-adapted evaporation estimates performed using ERA5 and met-station$_{SDH}$ during the period 2016-2020.* Evaporation monthly integration comparison metrics are between $P_{SDH}$ estimates using ERA5 and observation from met-station$_{SDH}$ (Table 1).

|  | RMSE | MAE | R | $R^2$ | m |
|---|---|---|---|---|---|
| Site-addapted Penman ($P_{SDH}$) | 73 W m$^{-2}$ | 17 W m$^{-2}$ | 0.88 | 0.78 | 0.98 |
| Standard Penman ($P_{stdr}$) | 149 W m$^{-2}$ | 88 W m$^{-2}$ | 0.87 | 0.76 | 1.35 |
| Radiative contribution to $L_vE$ | 94 W m$^{-2}$ | 28 W m$^{-2}$ | 0.85 | 0.73 | 1.02 |
| Aerodynamic contribution to $L_vE$ | 121 W m$^{-2}$ | 64 W m$^{-2}$ | 0.87 | 0.76 | 0.30 |
| $P_{SDH}$ E daily integration | 0.7 mm | 0.6 mm | - | - | - |
| $P_{SDH}$ E monthly integration* | 14.2 mm | 7.2 mm | 0.90 | 0.81 | 1.34 |

Secondly, the coefficient improves $L_vE$ estimations by mitigating the fluxes when the water in the lake is frozen in a factor of 0.3 (Appendix A4). Table 2 summarizes comparative statistical metrics between the results obtained using a standard and a site-adapted Penman equation with observations.

Finally, in Figure 4d, we integrate sub-diurnal evaporation estimates for validating our results during the entire E-DATA period. The Figure shows the daily evaporation between the EC observations and the $P_{SDH}$. Daily values show differences

of ~0.65 mm between observations and estimations (RMSE: 0.7 mm; MAE: 0.6 mm). Integrating the whole E-DATA period, the differences are ~5 mm: 38 mm for $P_{SDH}$ and 33 mm for $EC_{water}$. To place these differences into perspective, it is worth noting that our focus in this research is to study the climatology of the evaporation in this region. As such, we consider that mean daily errors below 1 mm per day are low enough to use Equation 1 using the ERA5 downscaled data for long-term actual evaporation estimations.

Nevertheless, to extend our validation into a longer period analysed in sections 3.2 and 3.3, Figure 5 shows the $L_vE$ calculated using two methods: (1) the site-adapted Penman monthly evaporation estimates using ERA5 downscaled data and (2) observations from the met-station$_{SDH}$ between 2016-2020. We find a good agreement between both estimates. The results show that ERA5 follows the seasonal cycle ($R^2$: 0.81) satisfactorily. However, ERA5 evaporation overestimates the observations by 7.6%, which is consistent with the overestimation that ERA5 reported evaporation results with respect to the EC

observations during the E-DATA period (6.1%).

The previous evaluation provides enough support to use the site-adapted Penman evaporation results to count with high-quality long-term (1950-2020) actual evaporation estimates at local (saline lake) scales and high time resolution (1-hour).





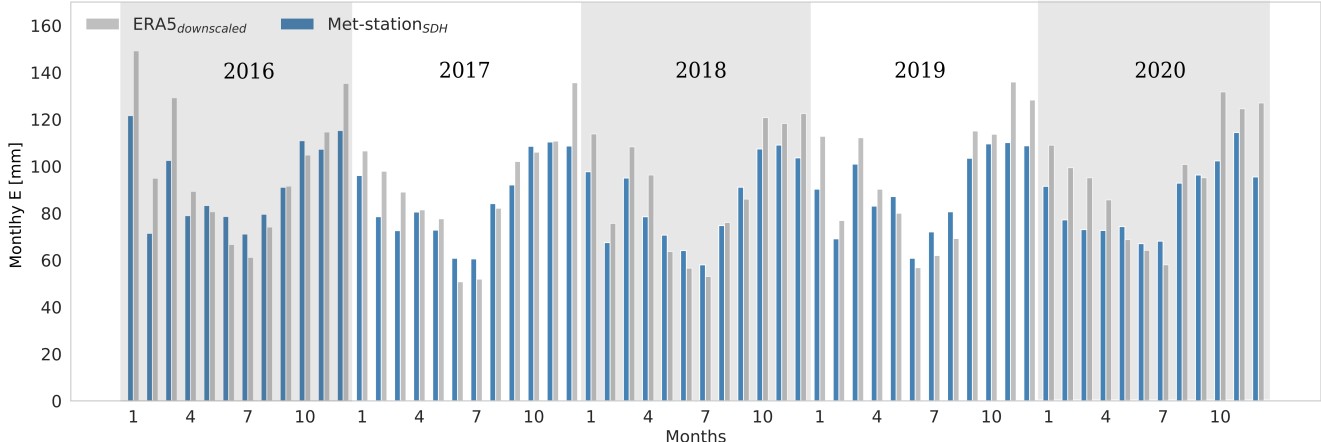

**Figure 5.** Monthly integrated evaporation obtained through the site-adapted Penman equation using ERA5 and met-station$_{SDH}$ standard meteorological data in the period 2016-2020.

## 3.2 Seasonal perspectives of evaporation and precipitation

Evaporation estimated sub diurnally through the Penman equation also presents significant seasonal changes that can have

high impacts on water resources. This section first analyzes the seasonal cycles of actual evaporation by describing the changes in its radiative and aerodynamic contributions. In addition, we include the precipitation in the analysis for being an essential component in the water balance. Secondly, we analyze the seasonal evaporation and precipitation impacts on the water balance of the saline lake of the Salar del Huasco.

### 3.2.1 Evaporation and precipitation seasonal cycles

Figure 6a shows the actual evaporation seasonal average from 1950 to 2020 over the saline lake of Salar del Huasco. In general, seasonal changes of evaporation show their highest monthly values (>90 mm) during austral summer (JFM) and spring (OND). Within these seasons, October, November, and December present the highest monthly evaporation (107-120 mm). Even though the summer also presents high monthly evaporation (90-107 mm), these months also show the highest variability (standard deviation of 13.5-16.5 mm). The variability observed during summer months is because of the rainy

season that usually extends over the summer (Vuille et al., 2000; Garreaud et al., 2003). Evaporation has its lowest rates during autumn and winter (<78 mm per month). Moreover, within these seasons, the months of June, July, and August show the lowest monthly evaporation (~50 mm) and the lowest variability of the year (standard deviation of 7 mm per month). On the other hand, the seasonal variability of precipitation is shown in Figure 6b. Precipitation in the Salar Huasco basin shows a very clear seasonal cycle, with the onset of the rainy season in late spring (ND) and the offset end of summer (MA). However, this

rainy season presents high variability over the years. The rest of the seasons show precipitation values below 25 mm per month, where June and July present a slightly higher variability.





**Figure 6.** Seasonal variability of monthly evaporation (a) and precipitation (b) rates in the period 1950-2020. The boxes represent the 25%-75% interquartile, the grey horizontal line is the median, the red dots are the mean, and the bars represent maximum and minimum values. Outliers have been removed and annual evaporation and precipitation have been calculated over the entire period.



To give a synoptic-scale perspective of the seasonal changes in local evaporation and precipitation presented in the saline lake of Salar del Huasco, Figure 7 shows the seasonally averaged moisture sources of the Altiplano region. Here, we quantify the regions where evaporation occurs which results in precipitation over the Altiplano region (grey box in Fig. 7). As most

precipitation occurs in austral summer (Fig. 6) the moisture sources are also largest in these seasons. We observe three principal moisture sources that contribute to precipitation in the Altiplano region during the year. The first one comes from the northeast (Amazon basin) and results from the veering of trade winds southwestwardly into the Andes mountains, associated with the continental low formed by the summery south-equator position of the Intertropical Convergence Zone, ITCZ (Aceituno, 1992). This southwestward flux is the most pronounced during the summer, transporting moisture ($\sim$50 mm) into the Altiplano region.

This marked moisture flux suddenly decreases towards the autumn and winter ($\sim$10 mm). During these seasons, the trade winds return to their normal westwardly direction (Figs. 7b and 7c), resulting in low moisture transport ($\sim$20 mm) into the region. Besides moisture transport into the region from the northeast, there is also recycling of moisture within the region, which can be considered to be a second moisture flux. Especially during summer, evaporation contributes to precipitation within the region, as can be seen by the high moisture source values around lake Salar del Huasco in JFM and OND (Figs. 7a and 7d). In

addition to the contributions from evaporation over land, there is also a positive moisture source from the Pacific Ocean (south-southwest). In the absence of precipitation over the ocean, we can assume that the evaporation over the ocean contributes to the precipitation over land in the Altiplano region. Finally, this third moisture flux is associated with the subtropical anticyclone and stratocumulus cloud deck (Rutllant et al., 2003; Lobos-Roco et al., 2018). This flux transports a very low but persistent amount of moisture into the Altiplano region ($<$5 mm) due to the steep topography presented on the western slope of the Andes

mountains, which in combination with the anticyclone, limits the eastward flow up to the mountains. Despite the coarse model resolution, this low moisture transport has been reported using high-resolution modeling and airborne observations by Suárez et al. (2020) and Lobos-Roco et al. (2021).

To unravel the processes involved in the seasonal evaporation, we analyze the seasonal variability of the drivers that control it. Figure 8 shows the seasonal cycle of the radiative and aerodynamic contribution of the Penman equation and subsequent

correlations with monthly evaporation rates.

Figure 8a shows the seasonality of the radiative contribution to evaporation, whose highest values correspond to the spring and summer and slowly decrease towards the winter only to increase again in early spring. The seasonality of the radiative contribution is similar to that of evaporation shown in Figure 6b, but it presents two distinctive characteristics. Firstly, from November to March, there is a larger scatter (standard deviation $>$ 12 W m$^{-2}$), where the radiative contribution to evaporation

can be high at 170 W m$^{-2}$ or low at 20 W m$^{-2}$. This large variability is directly related to the summer rainy season (Vuille et al., 2000), where the presence of clouds largely modulates the available net radiation (Houston, 2006). This double feedback that precipitation has over the radiation might explain the large scatter in the radiative contribution to evaporation during spring-summer. Secondly, the small variability (standard deviation of $\sim$3 W m$^{-2}$) of the radiative contribution during the winter months is related to the atmosphere's stability, characterized by the dry weather and cloudless conditions during most of this

period. Therefore, there is enough evidence to support the idea that radiative contribution controls evaporation at a seasonal scale (R$^2$ = 0.91, as shown in Fig. 8b).

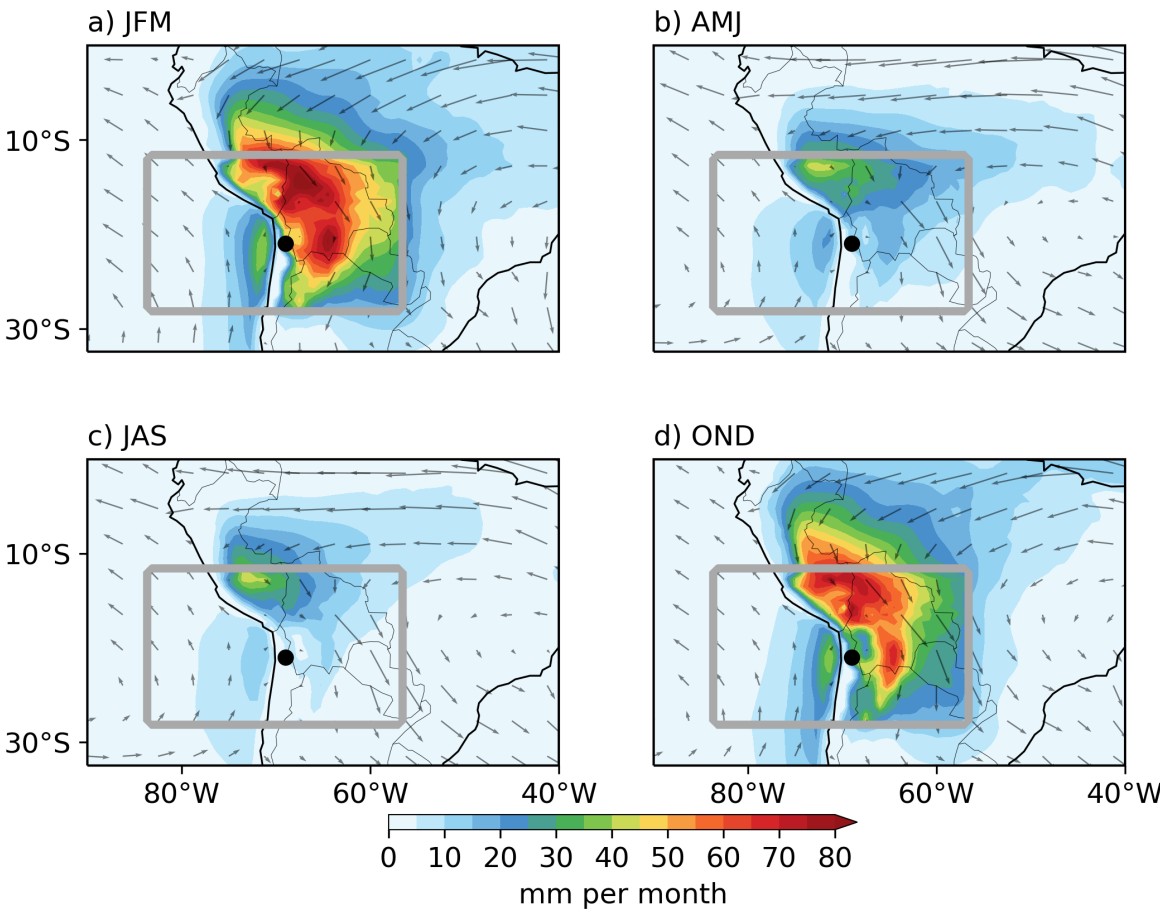

**Figure 7.** Seasonal variability of moisture sources (shaded) in mm per month and vertical integrated moisture transport (arrows) over the Altiplano region for (a) summer, (b) autumn, (c) winter, and (d) spring. The grey square frames the Altiplano regions and surroundings for which the sources are determined. The black dot indicates the Salar del Huasco location.

Figure 8c shows the seasonality of the aerodynamic contribution to evaporation, where the highest and lowest values are observed in early spring (SON) and during summer (JFM), respectively. The variability of the aerodynamic contribution is fairly constant during the whole year (standard deviation of ~4 W m$^{-2}$), which is related to the seasonality of the wind
circulation patterns (Falvey and Garreaud, 2005). The wind seasonality also explains the highest and lowest aerodynamic contribution to seasonal evaporation. For example, the thermal contrast between the Pacific Ocean and the Atacama Desert reaches its maximum in November, resulting in the strong regional atmospheric eastward flow, responsible for the onset of diurnal evaporation in the Salar del Huasco (Lobos-Roco et al., 2021). To the contrary, during summer, predominant westward regional circulation from the amazon basin counteracts the eastward regional flow (Garreaud et al., 2003), decreasing the



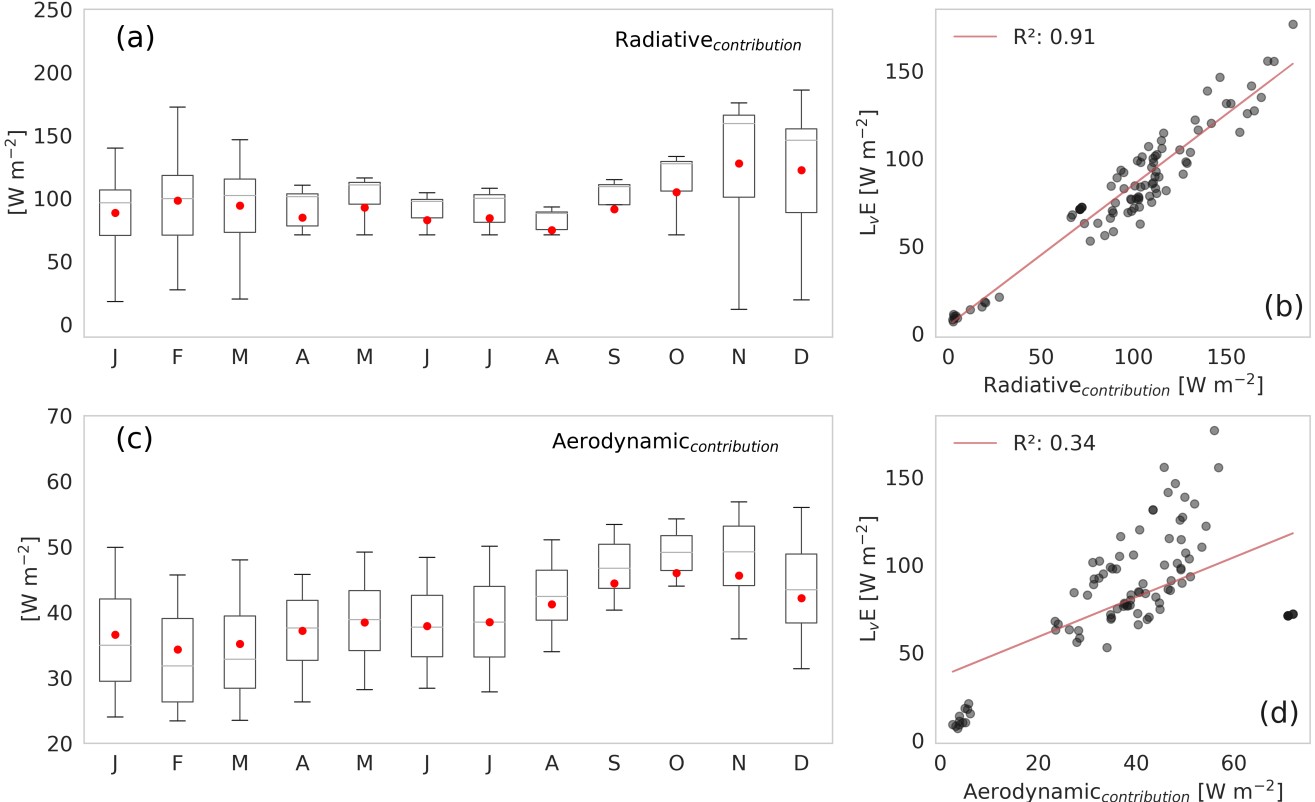

**Figure 8.** (a, c) Seasonal variability of radiative and aerodynamic contribution to evaporation during the period 1950-2020. The boxes represent 25%-75% interquantile, the grey horizontal line is the median, the red dots are the mean, and bars represent maximum and minimum values. Outliers have been removed. (b, d) Orthogonal regression of evaporation rates and its energy and aerodynamic contribution at averaged monthly scale.

wind speed (as described below). Finally, during winter, the lower thermal contrast between the Pacific Ocean and the Andes Altiplano, along with the absence of the summer westward regional flow, results in lower wind speed. Consequently, there is less aerodynamic contribution to evaporation. The scattered seasonality of the aerodynamic contribution to evaporation also results in a low correlation ($R^2 = 0.34$, as shown in Fig. 8d).

In summary, at the seasonal timescale, the radiative contribution term contributes significantly more to evaporation than the
aerodynamic term, representing 73% of the energy needed to evaporate the water from the saline lake in the Salar del Huasco. It is important to stress that mechanical turbulence (wind speed) is more relevant at the diurnal scales than available net radiation controlling evaporation (Lobos-Roco et al., 2021).





### 3.2.2 Seasonal changes in the saline lake water balance

To complete the seasonal analysis of evaporation in recent decades, we describe the spatial impacts of the evaporation-
precipitation variability on the saline lake of Salar del Huasco. Figure 9 shows the relationship between the spatial changes
of the saline lake and monthly evaporation and precipitation that occurred between 1985 and 2019. The seasonal variability
of the lake's area shown in Figure 9a reveals that the maximum extension (5 km$^2$) occurs during winter (JJA). During spring,
the lake's area decreases rapidly to its minimum extension ($\sim$1.3 km$^2$). The summer season shows high variability in the
lake's area (mean: $2 \pm 1.8$ km$^2$; mean value $\pm$ standard deviation). This variability is relatively constant towards winter and
decreases during spring, revealing that there is a significant interannual variation over the years, especially between March and
July, where precipitation is typically small (Fig. 6d). Thus, the increase in the lake's area is likely due to groundwater inputs
(Blin et al., 2022).

Regarding the relationship between the lake's area changes and evaporation, Figure 9b shows an orthogonal regression
between evaporation and lake extension changes. Here, we find a strong negative correlation (R$^2$ = 0.92), which reveals the
control that evaporation has over the lake discharge. The lowest evaporation rates (winter) coincide with the highest lake
extensions, and the highest evaporation rates (spring) coincide with the lowest lake surface. Regarding the relationship between
precipitation and lake's surface associated with the water recharge by precipitation, the relationship is indistinctive (Fig. 9a).
We find a high variability in the onset and offset of precipitation at the seasonal scale, from November to March. This high
variability in summer precipitation coincides with the larger variations in the lake area. As such, it is difficult to find a direct
relationship between precipitation and the lake's area. However, analyzing the means (solid lines Fig. 9a), we observe that
high precipitation rates do not directly impact the areal changes of the lake, which is reached 4 to 5 months after the rainy
season. For these reasons, the observations suggest that there is another process that modulates the lake recharge. Among the
alternatives that might explain the lake recharge, precipitation and groundwater input might play a role.

To unravel the role of ground water input into the Salar del Huasco lake, we perform a simple mass balance assuming a lake
depth between 0.05 and 0.20 m. Our lake mass balance results show that the monthly water required to represent the spatial
changes in the lake's surface are on the order of $\sim$345,000 m$^3$ per month ($\sim$0.1 m$^3$ s$^{-1}$). This estimation is reasonable as the
only stream flowing in the lake direction has an average flow of 0.13 m$^3$ s$^{-1}$ (Blin et al., 2022), which is measured about 1-2
km before the river water completely infiltrates into the ground. If one assumes that ERA5 precipitation is responsible of this
water flow, then a lake area of $\sim$13 km$^2$ is needed to explain it. When varying the water depth between 0.05 and 0.20 m, our
results changed less than 1%. As the mean observed lake's area is $\sim$2 km$^2$, groundwater is the water source that sustains this
habitat. This result agrees with the estimations performed by Blin et al. (2022). They quantified a flow in the range 0.14 to 0.2
m$^3$ s$^{-1}$ in the springs that discharge water into the lake, which is similar to the flow estimated in our work. It is important to
recall that our approach has important limitations. For instance, as the topography in the basin's sink is very flat, there is no
hypsometric curve that can relate the lake's volume as a function of depth. Also, the precipitation considered here corresponds
to that estimated in the lower part of the basin, whereas higher precipitation values occur at higher elevations in the basin
(Uribe et al., 2015; Blin et al., 2022). Hence, most of the groundwater recharge is expected to occur at higher elevations and/or





in locations where preferential flow exists, e.g., near the rivers of the basin. Then, this water will flow underground until it upwells into the lake. So, even though this approach has limitations, it allows for a first order approximation that can be used to understand the key components of the lake's water balance.



**Figure 9.** (a) Monthly mean variability of lake area, total evaporation, and total precipitation. Shades indicate the standard deviation of each variable. (b) Orthogonal regression between monthly lake area and monthly evaporation. (c) Orthogonal regression between monthly lake area and monthly precipitation.





### 3.3 Interannual perspectives of evaporation and precipitation

This section describes the interannual variabilities of evaporation and precipitation over the saline lake of the Salar del Huasco. First, we describe the climatological trends from 1950 to 2020. Secondly, we analyse the influence of ENSO and PDO global-scale phenomena on local evaporation and precipitation.

#### 3.3.1 Climatological trends of evaporation-precipitation

Evaporation trends in the saline lake of Salar del Huasco show an indubitable increase from 1950 to 2020. The rate of increase is about 2.1 mm per year (0.2 mm per month), with scattered interannual variability showing a significant increase. Figure 10a shows a 12-month moving average of monthly total evaporation. For 1950, monthly mean values are approximately 80 mm (950 mm per year), whereas in 2020, these values increased to ~100 mm (1,150 mm per year). The annual integrated evaporation rates averaged 1,075 mm (± 74 mm) with a minimum of 862 mm (1993) and a maximum of 1,210 mm (2010). This increase in evaporation has a correlation of 0.55 with air temperature (2 m), whose monthly averages increased 3 °C (0.04 °C per year), from 1950 to 2020 (Fig. 10a). Likewise, Figure 10b shows the precipitation trends in the area of the saline lake from 1950 to 2020. Total precipitation per year is set at 338 mm with a high variability of 248 mm per year. Precipitation shows an increasing trend in the last 70 years of 0.6 mm per year. Although this positive trend in precipitation is less significant than evaporation and presents more scatter, it is also in agreement with temperature increase.

#### 3.3.2 Influence of ENSO and PDO phenomena on evaporation-precipitation

Within this subsection, we analyze the influence of the ENSO and the Pacific Decadal Oscillation (PDO) on evaporation and precipitation variability.

Figure 11 shows the seasonal variability of monthly evaporation in the period 1950-2020 during cool (ONI < -0.5 °C), neutral (-0.5 °C < ONI < 0.5 °C) and warm (ONI > 0.5 °C) ENSO phases. In general, during cool ENSO phases, evaporation rates are 2% lower than those observed in the neutral ENSO phases, whereas during warm ENSO phases, evaporation is 15% higher than that observed in neutral ENSO phases. This variability becomes more significant from October to May, summer (JFM) being the season with the largest variability. During summer, evaporation under cool ENSO phases decreases by 4% with respect to neutral phases and increases 14% under warm conditions. Moreover, summer variability is the highest during warm ENSO phases, showing standard deviations of ~15 mm per month. The lowest evaporation variability occurs during the neutral phases, with standard deviations of 11 mm per month. In turn, during late autumn and winter seasons, the ENSO phenomenon influences the evaporation less in the saline lake of the Salar del Huasco since evaporation rate differences between cool, neutral and warm phases are lower than 2%. This analysis suggests that ENSO significantly influences evaporation during summer months, which is in line with other typical meteorological phenomena of the Atacama Desert, such as summer precipitation (Aceituno, 1988), and coastal fog formation (del Río et al., 2021).

The ENSO phases influence on evaporation observed at the seasonal scale is also present interannually. Figure 12 shows the relationship between ENSO phases and PDO phenomenon, with evaporation anomalies obtained using the site-adapted Penman





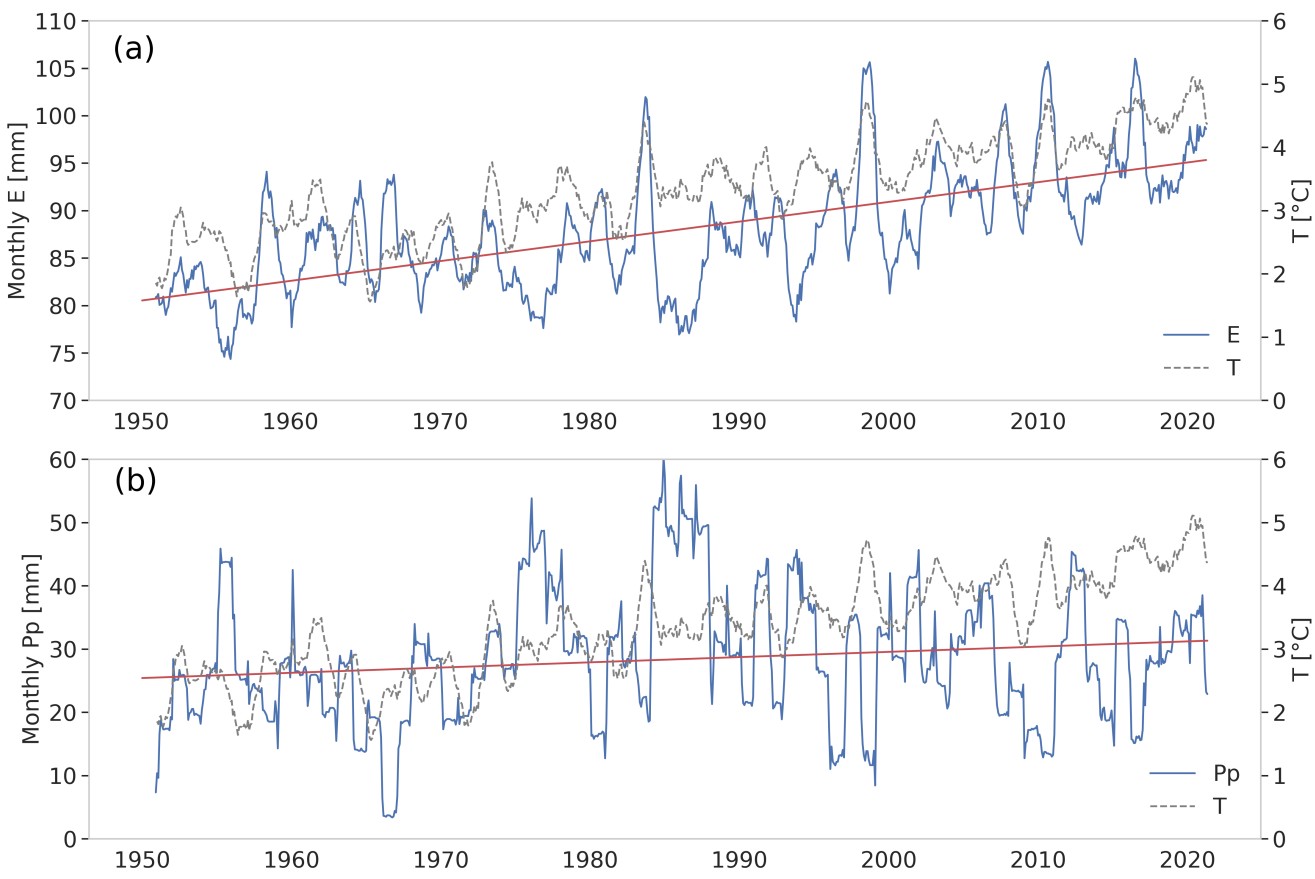

**Figure 10.** 12-month moving average monthly total evaporation (a) and precipitation (b), and 2-m mean air temperature over the saline lake of Salar del Huasco from 1950 to 2020. The evaporation trend line is depicted in red.

equation (Eq. 1) and downscaled ERA 5 data, and precipitation anomalies observed from ERA5 in the last seven decades in the shallow lake of the Salar del Huasco. Recall that ENSO is a recurrent phenomenon with an ill-defined periodicity, where in the last 70 years, 24% of the months have been influenced by warm ENSO phases and 26% by cool ones. However, the

frequency of this phenomenon is not constant, neither in intensity nor in time. The ONI varied between 0.5 and 2.6 °C during warm phases, and between -2 and -0.5 °C during cool phases, with a frequency between 2 and 10 years (Timmermann et al., 2018).

Figure 12a shows the 12-month moving average evaporation anomalies and the ENSO and PDO phenomena from 1950 to 2020. Positive monthly evaporation anomalies (> 5 mm) correlate with warm ENSO phases, whereas negative or non-

evaporation anomalies (< 0 mm) correlate with cool ENSO phases. The correlation between positive evaporation anomalies and warm ENSO phases is evident during the extreme ENSO events, i.e., events which occurred in 1983, 1997, and 2015. Likewise, the correlation between negative evaporation anomalies and cool ENSO phases is evident in 1988, 1998, and 2010.





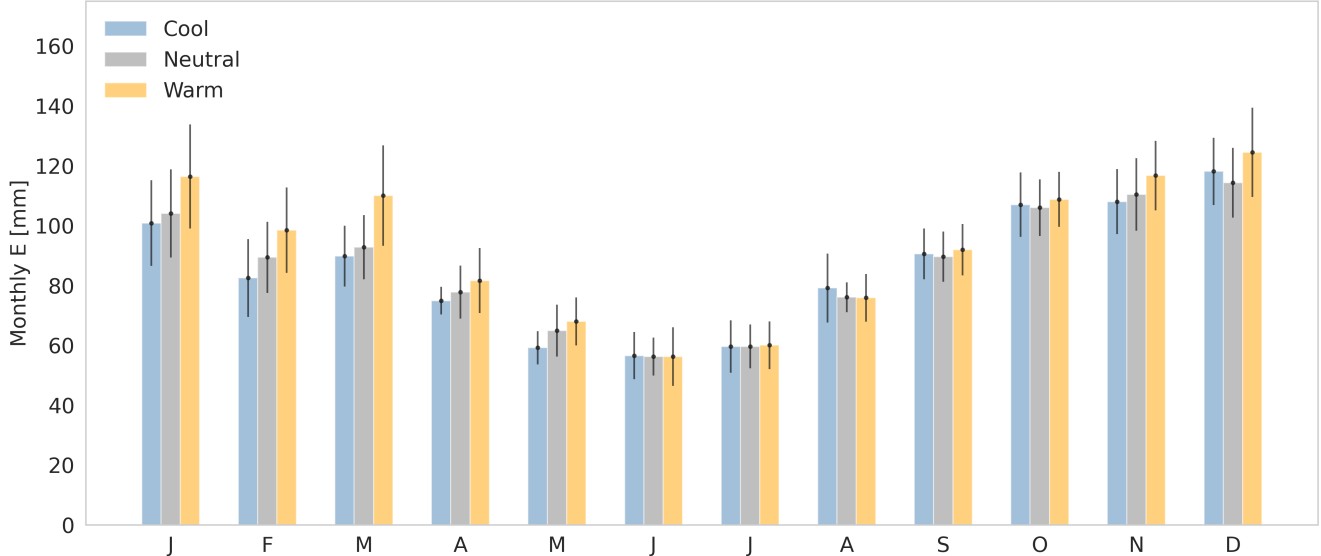

**Figure 11.** Interannual-seasonal variability of monthly evaporation in the period 1950-2020 separated by cool, neutral and warm ENSO phases. Error bars represent the standard deviation of every averaged month.

However, this trend is indistinct when monthly evaporation anomalies are close to 0 mm (e.g., in 1970, 1995, and 2001). Evaporation anomalies also have an interdecadal variability. For instance, between 1950 and 1975, negative evaporation anomalies

dominate. On the contrary, between 2000 and 2020, positive evaporation anomalies dominate, but only after a transition period that occurred between 1975 and 2000, where both positive and negative evaporation anomalies are present. Regarding larger macroclimatic phenomena, no significant correlation is found between PDO and evaporation anomalies.

The influence of the ENSO phenomenon also affects precipitation at Salar del Huasco. Figure 12b shows the 12-month moving average precipitation anomalies and the ENSO and PDO phenomena from 1950 to 2020. The influence of ENSO on

precipitation is opposite of that observed for evaporation. Here, positive monthly precipitation anomalies (> 5 mm) correlate with cool ENSO phases, whereas negative monthly precipitation anomalies (< 5 mm) correlate with warm ENSO phases. Contrary to evaporation anomalies, the relationship between precipitation and extreme ENSO events is indistinctive. For example, strong precipitation anomalies observed in 1985 disagree with an extremely cool ENSO phase. The same occurs for the extremely cool ENSO phase that occurred in 1999, where the precipitation anomaly is not correlated with high positive precip-

itation anomalies. However, the negative correlation trend between ENSO phases and precipitation anomalies is still evident. Precipitation anomalies also have an interdecadal variability that seems to be related to PDO anomalies. For example, between 1950 and 1970, there is a predominance of negative precipitation anomalies, which correlate with negative PDO indices. However, between 1970 and 2000, positive precipitation anomalies predominate along with positive PDO indices. Finally, between 2000 and 2020, negative precipitation anomalies predominate together with negative PDO indices. The negative relationship

**Figure 12.** (a) Monthly evaporation anomalies compared to ONI and PDO indices. (b) Monthly precipitation anomalies compared to ONI and PDO indices. Anomalies are calculated using the difference between the entire period mean, and 12-month moving averaged anomalies. Evaporation and precipitation anomalies are shown in colors, the ONI with a solid black line, highlighting the warm and cool phases, and PDO with a dashed line.





between precipitation and ENSO phases in the Altiplano region has also been reported by Aceituno (1988), Vuille et al. (2000), and Garreaud and Aceituno (2001).

**Figure 13.** Relationship between monthly evaporation and monthly precipitation anomalies between 1950 and 2020. The anomalies are classified into warm, neutral and cool ENSO phases. The symbol size reflects the ONI intensity, where ONI≥1 corresponds to an intense warm phase (circle) and ONI≤-1 to an intense cool phase (triangle).

To further quantify the opposing trend between evaporation and precipitation, Figure 13 shows the relationship between evaporation and precipitation anomalies categorized by ENSO phases. The trend between cool ENSO phases and negative evaporation anomalies is significant (∼-10 mm), although it is weaker during extremely cool phases. Likewise, the trend





between warm ENSO and positive evaporation anomalies is very clear, even during the most intense warm ENSO phases ($>$ 15 mm). Regarding precipitation anomalies, the trend shows a similar pattern, where negative precipitation anomalies ($\sim$-15 mm) are related to extremely warm ENSO phases, and highest positive precipitation anomalies are related to both cool and neutral ENSO phases.

Opposing behavior of ENSO influences on interannual evaporation and precipitation variability demonstrate the control that

global climate phenomena can exert at a local scale in the long term. As shown in Figure 10a, air temperature is strongly related to evaporation; thus, atmospherically warmer conditions in the Altiplano region during warm ENSO phases enhance evaporation. This warming intensifies the Pacific Anticyclone through the tropospheric thermal stratification (Falvey and Garreaud, 2009), resulting in cloudless conditions during summer of warm ENSO phases, i.e., an increase in the radiative contribution term of 17% as compared to the cool phases. Increased radiation also leads to an enhancement of the ocean-land thermal

contrast, enhancing the aerodynamic contribution to evaporation by 21% during warm ENSO phases in summer with respect to cool ENSO phases. This enhanced ocean-land thermal contrast also increases the atmospheric capacity to hold water vapor. Conversely, cool ENSO phases promote higher precipitation rates through the weakening of the Pacific Anticyclone, and the strengthening of the Bolivian low (Aceituno, 1988), which negatively affects the evaporation in two ways. First, the cloudy conditions that result from wet seasons inhibit the available energy required for evaporation (from 120 to 100 W m$^{-2}$), mainly

affecting the evaporation rates during the summer season (Fig. 11) but also interannual rates (Fig. 10a) (Houston, 2006). Second, during cool ENSO phases, a strong rainy season attenuates the characteristic regional atmospheric flow from the Pacific Ocean into the Andes (Lobos-Roco et al., 2021), significantly affecting the aerodynamic contribution to evaporation (Fig. 8b), decreasing it from 38 to 30 W m$^{-2}$.

## 4   Conclusions

We investigate the temporal changes of actual evaporation from sub-diurnal to climatological scales in a high-altitude saline lake ecosystem in the Atacama Desert. To this end, we combine observations of evaporation with two different model approaches. The first one downscaled ERA5 meteorological data (1950-2020) into local conditions observed at the saline lake of Salar del Huasco using artificial neural networks. The second one uses this downscaled data into a site-adapted Penman equation for open water evaporation. The intercomparison between our estimates and direct eddy-covariance evaporation mea-

surements, taken in a dedicated 10-days field experiment, shows a good sub-diurnal agreement (R$^2$: 0.78, m: 0.98) and errors $\sim$7% at diurnal and seasonal time scale.

Our first results reveal that ERA5/Penman successfully estimates open water bodies' actual evaporation from sub-diurnal to interannual scales. In analyzing the budget of evaporation at the sub-diurnal scale, the wind speed (aerodynamic contribution) is the main driver of evaporation, whereas, at the seasonal scale, the principal driver is the available energy (radiative

contribution).

Our findings show significant seasonal variations. Maximum rates are reached during spring (OND), minimum ones during winter (JAS), and a high variability is observed during summer (JFM). The seasonal changes in evaporation are explained





73% by the radiative contribution of the Penman equation, where seasonal changes in incoming radiation play a dominant role in the available energy for evaporation. In addition, our local estimates of evaporation and precipitation over the saline lake correlate with synoptic and seasonal variabilities of moisture transport. In analyzing this transport, we identify three main large-scale fluxes that contribute to the available moisture in the Altiplano region. The principal one transports a significant amount of moisture from the northeast (Amazon basin) and the humidity recycled from the evaporation-precipitation process during spring and summer. The third moisture flux identified transports a very low but persistent amount of moisture from the Pacific Ocean into the Atacama Desert consistently over the year. This moisture flux is strongly limited by the subtropical anticyclone and the steep topography. In addition, the seasonal variation in evaporation and precipitation along the analysed period impacts the saline lake. Our analysis suggests that evaporation is the principal driver of the lake discharge, explaining 92% of it. However, the recharge of the lake still remains unknown, since the role of precipitation continues elusive and hasn't yet been quantified. By analyzing the saline lake mass balance, we conclude that the water input required to explain the lake's spatial changes significantly exceeds that of precipitation. Therefore, we conclude that groundwater inputs play an essential role in the lake's recharge.

Evaporation also present a interannual variability, where the ENSO phenomenon plays an important role. Our results reveal that ENSO phases affect the evaporation rates during the summer: warm phases increase evaporation by 15% , whereas cool ones decrease it by 4%. Concerning the driving components of evaporation, radiation controls these interannual changes in summer. This control is given by the cloudy or cloudless conditions that characterize ENSO cool and warm phases, respectively. However, this is also explained by the aerodynamic contribution during the cold phases. The weakening of the Pacific Ocean anticyclone promotes the entrance of wet eastern flow that decreases the usual westerly flow, affecting the contribution of wind to evaporation. Analyzing the evaporation and precipitation anomalies compared to the Oscillation El Nino Index (ONI), we find that ENSO phases correlate positively to evaporation anomalies but negatively to precipitation ones. These correlations mean warm ENSO phases are mainly characterized by higher evaporation rates and cool phases with higher precipitation rates. In addition, climatological trends show that evaporation has increased by 2.1 mm per year during the entire study period according to global temperature increases.

Finally, our study gives a first multi-scale temporal approach to actual evaporation and its role in the water balance of the Atacama Desert. We demonstrate that long-term actual evaporation can be estimated reliably through a simple approach based on observations and reanalysis data. However, we acknowledge that longer-term actual evaporation measurements are needed to reduce the 7% uncertainty that the site-adapted Penman equation brings. Likewise, further research can be done on the site-adapted Penman equation coefficients to apply the same approach to different but more common surfaces of the desert (wet salt, wetlands, and sparse vegetation lands). Additionally, the interannual variability of evaporation-precipitation and moisture transport must be analysed by higher-resolution models to better understand the local impacts related to the sharp topography and land-use changes and the ENSO phenomenon.

*Data availability.* https://data.mendeley.com/datasets/c5s6zk2rmz/2





**Appendix A: Site-adapted Penman Equation**

Here, we introduce the radiative and aerodynamic contributions to the Penman (1948) equation. We also provide a physical meaning to the two coefficients used in the modified Penman equation: the coefficient to compensate for the absence of surface energy balance closure ($c_{EBNC}$) and the coefficient to account for the ice conditions ($c_{ice}$) above the saline lake of Salar del

Huasco. The modified Penman equation reads as:

$$L_v E = c_{ice} \frac{s}{s+\gamma} c_{EBNC} \overbrace{(R_n - G)}^{\text{Radiative}} + \frac{\rho_a c_p}{s+\gamma} \overbrace{\frac{1}{r_a}}^{\text{Aerodynamic}} (e_s - e), \qquad (A1)$$

where $s$ [Pa K$^{-1}$] is the slope of the saturated vapor pressure curve, $\gamma$ [Pa K$^{-1}$] is the psychrometric constant, $R_n$ [W m$^{-2}$] is the net radiation, $G$ [W m$^{-2}$] is the ground heat flux, $\rho$ [kg m$^{-3}$] is the dry air density, $c_p$ [J K$^{-1}$ kg$^{-1}$] is the air's specific heat at constant pressure, $r_a$ [s m$^{-1}$] is the aerodynamic resistance, $e_s$ [Pa] is the saturated vapor pressure, $T_a$ [K] is the air

temperature, and $e$ [Pa] is the vapor pressure at a measured level. Below, we detail the calculation and justification of each term in equation A1.

**A1  Radiative contribution**

The radiative contribution to the latent heat determined from Equation A1 depends on the available energy, i.e., $R_n - G$. Net radiation, $R_n$, is estimated as:

$$R_n = Sw_{in} - Sw_{out} + Lw_{in} - Lw_{out} = (1-\alpha)Sw_{in} + Lw_{in} - Lw_{out} \qquad (A2)$$

where $Sw_{in}$ [W m$^{-2}$] is the incoming shortwave radiation, which is provided by the ERA5 dataset (Hersbach et al., 2020); $Sw_{out}$ [W m$^{-2}$] is the outgoing shortwave radiation; $\alpha = 0.13$ [-] is the albedo, obtained during the E-DATA field campaign (Suárez et al., 2020; Lobos-Roco et al., 2021); $Lw_{in}$ [W m$^{-2}$] is the incoming longwave radiation; and $Lw_{out}$ [W m$^{-2}$] is the outgoing longwave radiation.

The $Lw_{in}$, which includes the cloud influence, is calculated using the model suggested by Sugita and Brutsaert (1993). This model corrects the clear-sky incoming longwave radiation ($Lw_{in,cs}$) calculated with the Stefan Bolztmann law, in the following way:

$$Lw_{in} = Lw_{in,cs}(1 + c_1 c_f^{c2}) = \sigma \epsilon T_a^4 (1 + c_1 c_f^{c2}) \qquad (A3)$$



where $\sigma$ is the Stefan Boltzmann constant ($5.67 \cdot 10^{-8}$ W m$^{-2}$ K$^{-4)}$); $\epsilon$=0.68 is the air emissivity, which is derived from
E-DATA measurements, and $T_a$ is the air temperature at 2-m height, obtained from ERA5 downscaled data; and are empirical
constants (Sugita and Brutsaert, 1993); and $c_f$ is the cloud factor proposed by Crawford and Duchon (1999):

$$c_f = 1 - \frac{Sw_{in}}{0.9c_s} \tag{A4}$$

Here, $c_s$ corresponds to the extraterrestrial incoming shortwave radiation multiplied by 0.9 to get the percentage of radiation
that reaches the surface at $\sim$4000 m asl empirically determined during the E-DATA experiment.

The $Lw_{out}$ is calculated using the methodology suggested by Holtslag and Van Ulden (1983):

$$Lw_{out} \equiv \sigma T_a^4 c_3 R_{n,ini} \tag{A5}$$

where $c_3 = 0.02$ is an empirical coefficient, and $R_{n,ini}$ corresponds to an initial value of net radiation, estimated as $0.76Sw_{in}$,
according to E-DATA observations (Suárez et al., 2020). Then, $R_{n,ini}$ is solved iteratively using the following expression:

$$R_{n,ini} = (1 - \alpha)Sw_{in} + Lw_{in} - Lw_{out} \tag{A6}$$

One iteration consists of solving Equation A5 using $R_{n,it}$. The value of $Lw_{out}$ is then used in Equation A6 for solving $R_{n,it}$.
Finally, this new value of $R_{n,it}$ is used again in Equation A6. After ten iterations, $Lw_{out}$ values do not change significantly.

The ground heat flux, $G$, which is required to estimate the available energy, is determined as a function of net radiation as:

$$G = c_4 R_n \tag{A7}$$

where $c_4 = 0.25$ corresponds to an empirical coefficient based on the $R_n/G$ ratio observed during the E-DATA experiment
for $Sw_{in} > 50$ W m$^{-2}$.

Figure A1 shows an orthogonal regression estimated through this model and Rn-obs observed over the saline lake, which
validates the net radiation estimated by the model (see equation A2).

**A2   Aerodynamic contribution**

To calculate the aerodynamic term (Eq. A1), we use $T_a$, specific humidity, $q$, and wind speed at 2 m, $U$, from the ERA5
downscaled dataset. We parametrize the aerodynamic resistance term, $r_a$, by prescribing values for the two wind regimes
observed by Lobos-Roco et al. (2021). Figure A2 shows the prescribed values for $r_a$ , being $r_a$= 60 s m$^{-1}$ for $U > 3$ m s$^{-1}$
(windy regime during the afternoon) and $r_a$= 250 s m$^{-1}$ for $U < 3$ m s$^{-1}$ (calm regime during the morning). This prescription
is given by the rapid change of $r_a$ in the transition of the two diurnal wind regimes, where these values are representative.
It is important to stress two aspects that justify this prescription. Firstly, there are no significant changes in the aerodynamic





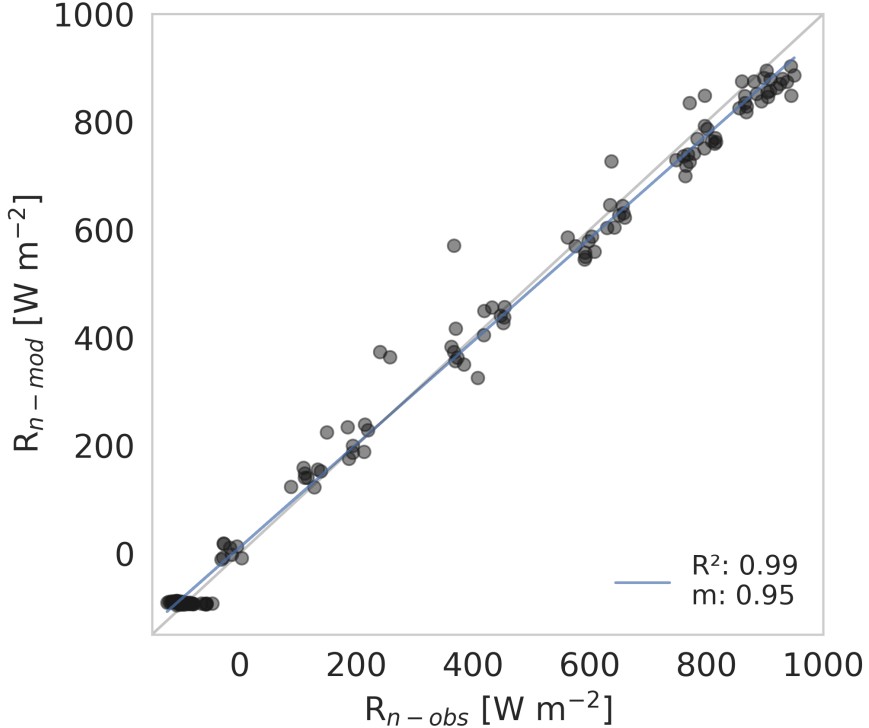

**Figure A1.** Orthogonal regression between Rn observed in the E-DATA field campaign and that modeled using equation A1.

contribution term of Equation A1 when $r_a > 200$ s m$^{-1}$. For this reason, we decide to use a wind regime averaged value. Secondly, the main idea behind estimating evaporation through Equation A1 is to use standard meteorological data readily available in a simple way.

The saturated vapor pressure, $e_s$, which is also required in the aerodynamic contribution term, is approximated using the August–Roche–Magnus equation (Moene and Van Dam, 2014):

$$e_{sat}(T_{ak}) = 611 exp\left[\frac{a(T_{ak} - 273.15)}{-b + T_{ak}}\right] \tag{A8}$$

where $T_{ak}$ [K] is the absolute air temperature obtained from the ERA5 downscaled data, and $a$ and $b$ are 17.625 and -30.03, respectively.

## A3  Energy balance non-closure coefficient

Since the Penman equation assumes a perfect energy balance closure, and the E-DATA field data show a significant energy imbalance (Suárez et al., 2020), we introduce the energy balance non-closure coefficient, $c_{EBNC}$. Hence, this coefficient corrects the available energy to improve the energy balance closure. We observe two different non-closure balances that depend





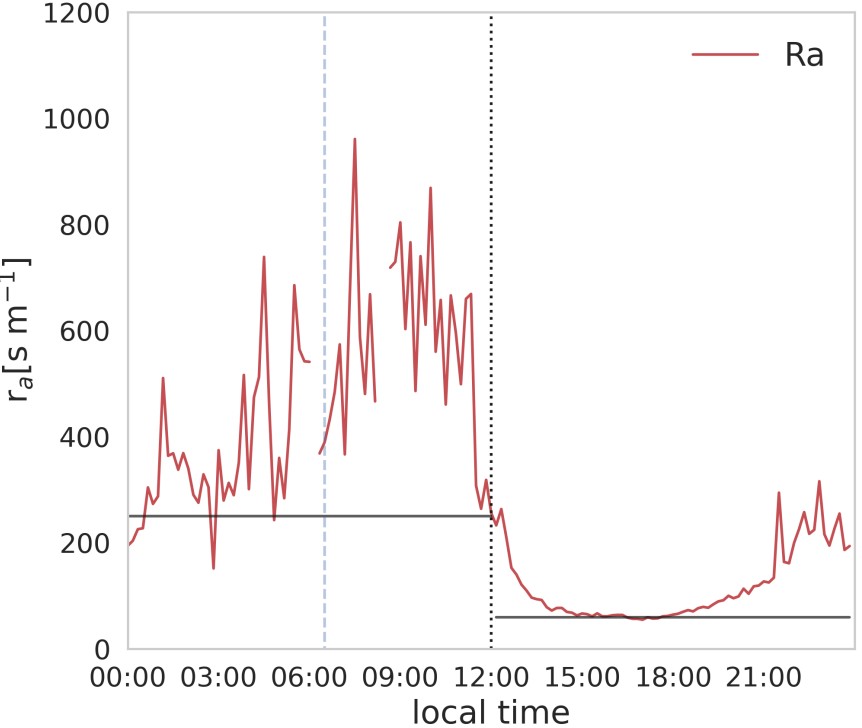

**Figure A2.** Diurnal averaged aerodynamic resistance observed above the water surface during the E-DATA, and the black lines are the prescribed values under calm and windy regimes.

on the wind regime (Fig. A3). Therefore, we set $c_{EBNC} = 0.3$ for $U < 0.3$ m s$^{-1}$ (calm regime during the morning) and $c_{EBNC}$ $= 0.7$ for $U > 3$ m s$^{-1}$ (windy regime during the afternoon).

## A4 Ice coefficient

Ice formation significantly restricts evaporation because it isolates the water from the atmosphere below a thin ice cover. Then, in the absence of wind, the available energy is used first to melt the ice before water evaporation occurs. Vergara-Alvarado (2017) demonstrated that a ∼3-5 cm thick ice cover in the Salar del Huasco saline lake reduced the turbulent fluxes to zero by creating an isolating layer between the water surface and the atmosphere. Thus, neglecting ice formation leads to an overestimation of the latent heat flux. A complete ice model requires the derivation of heat transfer fluxes or an elaborated

parameterization using variables and parameters that usually are not available in standard meteorological datasets (Echeverría et al., 2020). For this reason, we use an ice coefficient, $c_{ice}$, which ranges between 0 and 1, that is related to the number of hours under freezing conditions. We assumed that ice melting occurs when air temperatures are below 270 K. Table A1 shows





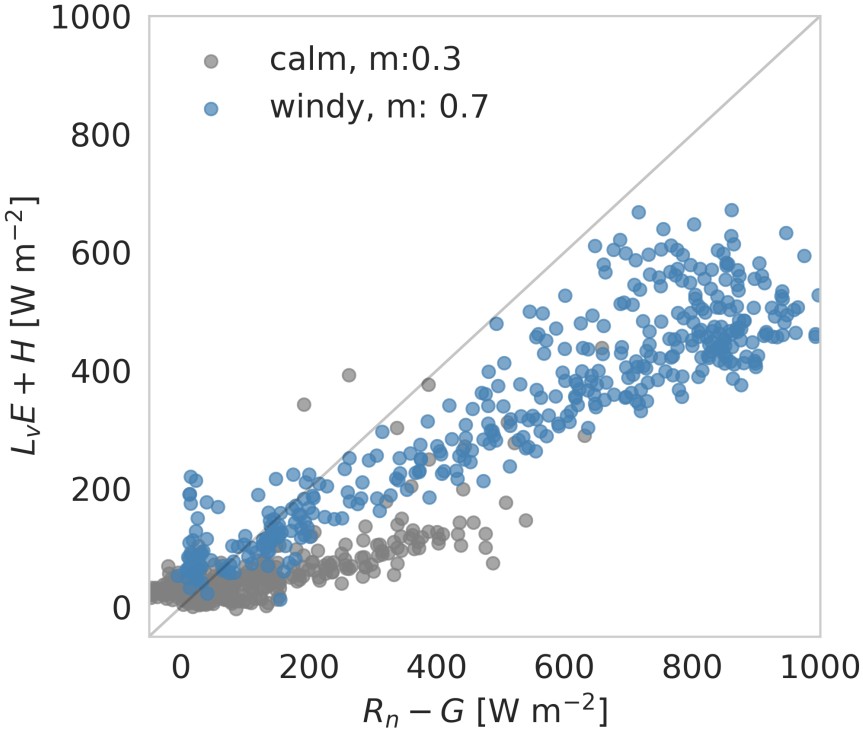

**Figure A3.** Surface energy balance observed at the water surface during the E-DATA under calm and windy regimes.

**Table A1.** Categorization of freezing hours and the ice coefficient.

| Freezing hours (FH) | Ice coefficient $c_{ice}$ |
|:---:|:---:|
| 8 > FH | 0.30 |
| 4 < FH < 8 for day | 0.40 |
| 4 < FH < 8 for night | 0.78 |
| FH < 4 | 1.0 |

the categorization of the freezing hours (FH) and the corresponding $c_{ice}$. Figure A4 shows the effect that the ice coefficient has in estimating latent heat flux during freezing days together with a time series of the freezing hours during the E-DATA.

*Author contributions.* The article was written by F. Lobos-Roco with the assistance of Dr. O. Hartogensis, Prof. Dr. J. Vilà-Guerau de Arellano and Dr. F. Suárez. The data were analysed by F. Lobos-Roco and Dr. F. Suarez, who mainly contributed to ANN data processing.





**Figure A4.** (a) The effect of the ice coefficient into the site-adapted Penman evaporation estimates. (b) Degree hours and air temperature time series.





Ms. Ariadna Huerta collaborated in Equation 1 (Appendix A); Dr. Imme Benedict collaborated in section 3.2.1 (Fig. 7); Dr. Alberto de la Fuente provided the data used in section 3.2.2 (Fig. 9).

*Competing interests.* The authors declare that they have no conflict of interest

*Acknowledgements.* This research received financial support from the Chilean National Commission of Science and Technology through the project ANID/FONDECYT/1210221. Support for Felipe Lobos-Roco was provided by the Wageningen University Ph.D. Sandwich Project no.: 5160957644. Dr. F. Suárez acknowledges support from the Centro de Desarrollo Urbano Sustentable (CEDEUS - ANID/FONDAP/15110020) and from the Centro de Excelencia en Geotermia de los Andes (CEGA - ANID/FONDAP/15090013). Finally, we acknowledge Robin Palmer (English editing) and the two anonymous reviewers for their valuable contributions to this manuscript.



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
