# Peer review of "Multi-scale temporal analysis of evaporation on a saline lake in the Atacama Desert"

_Hydrology and Earth System Sciences, 2022_

## Referee Comment (RC1)

The manuscript entitled "Multi-scale temporal analysis of evaporation on a saline lake in the Atacama Desert" by Felipe Lobos-Roco, O. Hartogensis, F. Suárez, A. Huerta-Viso, I. Benedict, A. de la Fuente and J. Vilà-Guerau de Arellano explores the magnitude and driving processes of evaporation from the Salar del Huasco, an endorheic salt flat located on the Altiplano Plateau, on different temporal scales, combining local observations, ERA5 reanalysis and different model approaches. The authors show that drivers of evaporation change from sub-diurnal scale (wind turbulence), to seasonal (net radiation) and interannual scale (global temperature, ENSO). Further, the authors assessed the lake water balance, demonstrating that evaporation controls lake level changes and groundwater recharge is significant. The results of this study are of high significance for water resource management in the context of industrial use and climate change, as well as for protection of biodiversity.

I would recommend moderate revisions to improve the presentation of the manuscript. Key issues need to be addressed as follows.

The authors precisely and clearly describe the performed experiments, datasets, calculations and obtained results. I acknowledge that the authors attempt to better guide the reader through the number of applied methods and obtained results by introductory paragraphs to each subsection. However, by this, many information are repeated several times, making the text lengthy. The information in these introductory paragraphs becomes most times already clear from the titles of the subsections. Thus, I suggest to remove these lines to make the text more concise and go straight to the point. I specify the respective lines below in the line-by-line comments. In general, the very detailed description and the number of applied analyses makes it, however, difficult for the reader to identify key messages. When revising the manuscript, the authors should check for redundancies of information and try to be more concise. The authors may also consider to separate results (including downscaling, variability of evaporation on sub-diurnal, seasonal and interannual scales, and precipitation moisture sources) from the discussion (comparison/discussion of driving processes of evaporation on different timescales, lake water balance, large-scale implications (see comment below)).

The authors estimate the monthly water loss from the lake based on their evaporation estimates and lake area. Can the authors also estimate the amount of groundwater recharge based on the seasonal changes in the lake surface area and evaporation, and/or show if there is seasonal variability in the groundwater influx? Or is this beyond the limitations of the approach? As the precipitation amount is low and less variable in the autumn/winter months, the groundwater recharge may be estimated with sufficient precision. Also, the authors do not consider runoff from the catchment caused by precipitation, which may increase the contribution of precipitation (directly + indirectly) significantly. When taking runoff into account, may the amount of groundwater recharge necessary be reduced?

The discussion of the results in a broader context is limited. To which extent are the obtained results applicable to other salar systems in the Altiplano Plateau? Which implications have the increase in evaporation rate / decrease in precipitation for saline lakes in the Atacama Desert and water resource management in this region? I would acknowledge if the authors can add a short paragraph on this subject, at least as part of the outlook in the Conclusion.

Line-by-line comments:

Line 31: "[…] within the Atacama Desert where rainfall provides a source of water for northern Chile" => sense of the phrase unclear; In the Atacama Desert, precipitation is generally scarce and groundwater, rather than precipitation, the major water source.

Line 38 ff.: The authors refer here mainly to previous studies at the Salar del Huasco done by the authors of this study. However, there are a number of other studies in recent years at the Salar del Huasco, which investigated, for example, the groundwater regime (e.g. Johnson et al., 2010; Jayne et al., 2016; Scheihing et al., 2017), or the hydrological functioning of the salar (Voigt et al., 2021). The authors may consider to mention them too as the results of the present study are of major relevance for them.

Line 64/65: consider specifying the sentence to "This dependence of precipitation on climatic factors implies that […].

Line 88-102: The first and the second half of this paragraph contain kind of similar information. I recommend to merge them to avoid redundancies.

Line 107: With 135 km distance and ~4 km asl the Salar del Huasco is not really close to the ocean. It's rather the ocean-land thermal contrast that forces the atmospheric flow from the Pacific to the Altiplano in the afternoon hours.

Line 173: add that $EC_{water}$ takes measurements "above the saline lake" in contrast to met-station$_{SDH}$, which takes measurements above bare soil.

Line 187: As stated here, freezing is an important variable as it leads to a reduction of evaporation. It would be great if the authors can add some information on the periods of freezing at the Salar del Huasco and seasonal differences (e.g., how many days per season).

Line 203/204: May be combined with the first sentence of the paragraph to avoid redundancies.

Line 210: suggestion: Seasonal averages of moisture sources are "evaluated" rather than "shown"

Line 212: suggestion: "long-term water balance" rather than "mass balance"

Line 216: The lake's area estimates are obtained from de la Fuente et al. (2021)? Add reference.

Line 227-232: I think this paragraph is redundant. There is no need for an introductory paragraph That results obtained on i) diurnal, ii) seasonal, and iii) interannual scales are subsequently presents becomes already clear from the titles of the subsections.

Line 234-236: redundant. Go straight to the point. "Fig. 4 shows …"

Line 308: The authors highlight that local evaporation provides a significant moisture source for precipitation in the rainy season. Does this imply that evaporation forces precipitation, leading to the positive correlation between both (both high in austral summer, low in winter)? How may this be interpreted in terms of the anti-correlation between evaporation and precipitation observed on interannual scales?

Line 368: In the sentences before, the authors argue against precipitation as the main driver of lake recharge. Thus, only groundwater should remain as an alternative explanation.

Line 386-388: As mentioned before, for me, there is no need for an introductory paragraph to each section. The authors may consider to remove these lines.

Line 401/402: Redundant, see comment before. The authors may consider to remove these lines.

Line 482/483: doubling the information that seasonal changes are dominated by radiation (mentioned in line 479/480).

Line 503-505: Consider to merge the two sentences.

Line 587: ice melting occurs when air temperatures are "above" 270K rather than "below"?

---

## Author Comment (AC1)

Reviewer #1

We thank the reviewer for his/her constructive comments and valuable contribution to our manuscript. Below, we answered the line-by-line comments in blue font and indicated if, how, and where we introduced changes in the revised manuscript in a colored font.

The manuscript entitled "Multi-scale temporal analysis of evaporation on a saline lake in the Atacama Desert" by Felipe Lobos-Roco, O. Hartogensis, F. Suárez, A. Huerta-Viso, I. Benedict, A. de la Fuente and J. Vilà-Guerau de Arellano explores the magnitude and driving processes of evaporation from the Salar del Huasco, an endorheic salt flat located on the Altiplano Plateau, on different temporal scales, combining local observations, ERA5 reanalysis and different model approaches. The authors show that drivers of evaporation change from sub-diurnal scale (wind turbulence), to seasonal (net radiation) and interannual scale (global temperature, ENSO). Further, the authors assessed the lake water balance, demonstrating that evaporation controls lake level changes and groundwater recharge is significant. The results of this study are of high significance for water resource management in the context of industrial use and climate change, as well as for protection of biodiversity. I would recommend moderate revisions to improve the presentation of the manuscript. Key issues need to be addressed as follows.

The authors precisely and clearly describe the performed experiments, datasets, calculations and obtained results. I acknowledge that the authors attempt to better guide the reader through the number of applied methods and obtained results by introductory paragraphs to each subsection. However, by this, many information are repeated several times, making the text lengthy. The information in these introductory paragraphs becomes most times already clear from the titles of the subsections. Thus, I suggest to remove these lines to make the text more concise and go straight to the point. I specify the respective lines below in the line-by-line comments. In general, the very detailed description and the number of applied analyses makes it, however, difficult for the reader to identify key messages. When revising the manuscript, the authors should check for redundancies of information and try to be more concise. The authors may also consider to separate results (including downscaling, variability of evaporation on sub-diurnal, seasonal and interannual scales, and precipitation moisture sources) from the discussion (comparison/discussion of driving processes of evaporation on different timescales, lake water balance, large-scale implications (see comment below)).

**Answer:** We made an effort to reduce redundancies in the text as suggested by the reviewer. The changes are detailed in the line-by-line reply below. However, we opted to keep the results and discussions in one single section. The discussion we follow focuses on one spatial or temporal scale at a time and we have little discussion points that cross scales. Splitting the results and discussions in separate sections would result in a discussion section with the same sub-sections for the temporal scales with potentially more redundancy.

The authors estimate the monthly water loss from the lake based on their evaporation estimates and lake area. Can the authors also estimate the amount of groundwater recharge based on the seasonal changes in the lake surface area and evaporation, and/or show if there is seasonal variability in the groundwater influx? Or is this beyond the limitations of the approach? As the precipitation amount is low and less variable in the autumn/winter months, the groundwater recharge may be estimated with sufficient precision. Also, the authors do not consider runoff from the catchment caused by precipitation, which may increase the contribution of precipitation (directly + indirectly) significantly. When taking runoff into account, may the amount of groundwater recharge necessary be reduced?

**Answer:** Our approach, was a first-order approximation on a decadal timescale that omits groundwater recharge values at shorter timescales. Our estimations agree in the order of magnitude with long-term river flow observations in the basin that are obtained just before the river completely infiltrates to then travel as groundwater into the basin's sink (e.g., see Uribe et al., 2015 or Blin et al., 2022). On seasonal timescales, calculating groundwater variability would bring high uncertainty since this groundwater

also contributes to other surfaces like wet-salt and wetlands, in which seasonal variability of evaporation is different from the one present on the lake. For these reasons, we believe that more precise information is needed to reproduce the seasonal variability of groundwater flow, and thus this is beyond the limitation of our current approach.

We do not consider runoff from the catchment because there has not been a surface connection between the runoff and the lake in recent decades. This behavior occurs due to the highly permeable soil in the lake surroundings that infiltrates all possible surface runoff. Therefore, the contribution of runoff is negligible.

Action taken: we have introduced the following sentences to clarify this doubts in the manuscript:

Line 223:"Because most of the time there are no surface water inputs (negligible surface runoff is observed)"

Line 225:"However, we believe that more precise information is needed to reproduce the seasonal variability of groundwater flow."

The discussion of the results in a broader context is limited. To which extent are the obtained results applicable to other salar systems in the Altiplano Plateau? Which implications have the increase in evaporation rate / decrease in precipitation for saline lakes in the Atacama Desert and water resource management in this region? I would acknowledge if the authors can add a short paragraph on this subject, at least as part of the outlook in the Conclusion.

Answer: we agree to add this discussion as a part of the outlook in the conclusions.

Action taken: we have modified the last paragraph of the conclusion adding this discussion as:

"Finally, our study gives a first multi-scale temporal approach to understand actual evaporation, its role in the water balance of the saline lakes of Atacama Desert, under a context of climate change. We demonstrate that long-term actual evaporation is estimated reliably through a simple approach that combines observations and reanalysis data. However, we acknowledge the need of longer-term actual evaporation measurements to reduce the 7% uncertainty that the site-adapted Penman equation brings. Our approach aims at improving water resources management in arid regions. To generalise our approach further research will be needed on the site-adapted Penman equation coefficients for other surfaces in the Atacama Desert (wet-salt, wetlands, and sparse vegetation lands), as well as other arid regions worldwide. Moreover, the interannual variability of evaporation-precipitation and moisture transport must be analysed using higher-spatial-resolution models that include better the local impacts related to the sharp topography and land-use changes, as well as the ENSO phenomenon. Last, our site-adapted Penman approach must be corroborated in basins and lakes with different spatial scales, topography, and locations. "

**Line-by-line comments:**

Line 31: "[…] within the Atacama Desert where rainfall provides a source of water for northern Chile" => sense of the phrase unclear; In the Atacama Desert, precipitation is generally scarce and groundwater, rather than precipitation, the major water source.

Answer: comment accepted.

Action taken: we have modified the sentence to: "The Altiplano region has a unique environmental, economic, and social value due to its location within the Atacama Desert, where groundwater fed a short, annual rain-period provides the main source of water for northern Chile"

Line 38 ff.: The authors refer here mainly to previous studies at the Salar del Huasco done by the authors of this study. However, there are a number of other studies in recent years at the Salar del Huasco, which investigated, for example, the groundwater regime (e.g. Johnson et al., 2010; Jayne et al., 2016; Scheihing et al., 2017), or the hydrological functioning of the salar (Voigt et al., 2021). The authors may consider to mention them too as the results of the present study are of major relevance for them.

Answer: comment accepted.

Action taken: we have added the suggested references to the Introduction (see colored track-and-trace changes in the ms).

Line 64/65: consider specifying the sentence to "This dependence of precipitation on climatic factors implies that […].

Answer: comment accepted.

Action taken: we have modified the sentence as follows: "The ENSO influence on climatic factors such as precipitation implies that evaporation, as a temperature-dependent process, might also be affected by this phenomenon (Houston, 2006)."

Line 88-102: The first and the second half of this paragraph contain kind of similar information. I recommend to merge them to avoid redundancies.

Answer: comment accepted.

Action taken: We have removed the text from line 93 to 102, and modified it from line 88 to 93 as follows:

"In this study, we applied climatologically robust, downscaled reanalysis data to the saline lake of the Salar del Huasco. Although we focused on one particular saline lake, this kind of surface represents the main evaporation pathway of the Altiplano region (Houston, 2006). We hypothesized that the evaporation of the saline lake can be represented using an adapted version of the Penman (1948) equation. Confirmation of this hypothesis enables us to extend the adapted Penman model to the entire climatological period (1950-2020) and to investigate evaporation fluctuations and their drivers at seasonal and interannual scales. "

Line 107: With 135 km distance and ~4 km asl the Salar del Huasco is not really close to the ocean. It's rather the ocean-land thermal contrast that forces the atmospheric flow from the Pacific to the Altiplano in the afternoon hours.

Answer: comment accepted.

Action taken: We have modified the sentence as follows: "This endorheic basin is located to the west of the Andes, 135 km inland from the Pacific Ocean, and is subject to an intense and recurrent afternoon atmospheric flow from the ocean that transports relatively cold and humid air into the Altiplano (Lobos-Roco et al., 2021)".

Line 173: add that EC water takes measurements "above the saline lake" in contrast to met-station SDH , which takes measurements above bare soil.

**Answer:** comment accepted.

**Action taken:** the sentence was modified as follows: "The differences are related to the surfaces above which the instruments are installed, i.e. $EC_{water}$ above the saline lake and met-station$_{SDH}$ above bare soil (Fig. 1)".

Line 187: As stated here, freezing is an important variable as it leads to a reduction of evaporation. It would be great if the authors can add some information on the periods of freezing at the Salar del Huasco and seasonal differences (e.g., how many days per season).

**Answer:** comment accepted.

**Action taken:** we have calculated the freezing periods and added this information to Appendix A4 (line 586-587) as: "For this reason, we use an ice coefficient, $c_{ice}$, which ranges between 0 and 1, depending on the number of freezing hours per day (Table 1). The days are taken from midday to midday to include the night. The idea is that ice produced over longer periods takes longer to melt. We assumed that freezing occurs when the 2 m air temperature is below 270 K, slightly below the freezing temperature of clean water to include the effect of salinity. Based on this criterion, freezing days are distributed over the year as 6% in summer, 21% in fall, 41% in winter, and 31% in spring."

Line 203/204: May be combined with the first sentence of the paragraph to avoid redundancies.

**Answer:** comment accepted.

**Action taken:** we merged the sentence with the previous one as follows: "To get an overview of the moisture transport that results in precipitation over the Altiplano region and surrounding areas, we determine the moisture sources of a selected region spanning from 83° W to 57° E, and from 11° N to 27° S (Fig. 7)."

Line 210: suggestion: Seasonal averages of moisture sources are "evaluated" rather than "shown"

**Answer:** comment accepted.

**Action taken:** applied change as suggested

Line 212: suggestion: "long-term water balance" rather than "mass balance"

**Answer:** comment accepted.

**Action taken:** applied change as suggested

Line 216: The lake's area estimates are obtained from de la Fuente et al. (2021)? Add reference.

**Answer:** comment accepted.

**Action taken:** the reference de la Fuente et al. (2021) has been added.

Line 227-232: I think this paragraph is redundant. There is no need for an introductory paragraph That results obtained on i) diurnal, ii) seasonal, and iii) interannual scales are subsequently presents becomes already clear from the titles of the subsections.

Answer: We understand the reviewer's concern about some redundancies throughout the manuscript, and we have accepted most of her/his suggestions on this issue. However, for this specific paragraph, we opted to keep it. We think that the paragraph introduces well how results and discussions are structured, which helps the reader to navigate through this long section.

Action taken: For reducing the text length we have shortened the introductory paragraph as follows: "This section describes the diurnal, seasonal, and interannual variability of evaporation at the saline lake of Salar del Huasco. First, we analyse the diurnal variability of evaporation through the site-adapted Penman equation. Secondly, we analyze the seasonal variations of evaporation, its main drivers, and the role of evaporation in the water balance of the saline lake. Finally, we close the article by studying the climatological trends of evaporation-precipitation and the influence of the ENSO and PDO phenomena on their anomalies.".

Line 234-236: redundant. Go straight to the point. "Fig. 4 shows …"

Answer: comment accepted.

Action taken: we have removed this paragraph.

Line 308: The authors highlight that local evaporation provides a significant moisture source for precipitation in the rainy season. Does this imply that evaporation forces precipitation, leading to the positive correlation between both (both high in austral summer, low in winter)? How may this be interpreted in terms of the anti-correlation between evaporation and precipitation observed on interannual scales?

Answer: The moisture tracking model is based on input data at coarse resolution (1.5°), which is unable to capture the local evaporation observations/estimations that occur under the localized conditions such as those at the Salar del Huasco. Indeed, the moisture sources analysis shows that during the rainy season (summer) there is an increased contribution from local evaporation to the total moisture due to increasing temperatures, radiation and soil moisture (rain season). It should be noted that an increase in moisture sources within the domain, does not necessarily mean that precipitation originates from evaporation in the same grid cell.
Also, during summer there is still moisture transport from outside the selected region, as can be seen in Figure 7. It is well-studied that moisture in summer is transported from the Amazon basin into the Altiplano region, and its correlation with ENSO is highly variable in space (Aceituno, 1988). Our intention using the tracking model was to reinforce the idea that local seasonal changes of evaporation are also observed at larger spatial scale. To further understand the exact correlation between evaporation, precipitation and moisture sources at the scale of the Salar del Huasco, we should apply moisture tracking     model simulations on a finer grid.
In the conclusions section we state that higher-resolution models (compared to that used in this work) should be used to better understand the local impacts related to the sharp gradients in topography and land use.

Action taken: no action was taken

Line 368: In the sentences before, the authors argue against precipitation as the main driver of lake recharge. Thus, only groundwater should remain as an alternative explanation.

**Answer:** comment accepted.

**Action taken:** we modified line 368 as follow: "Thus, the alternative that explains lake recharge is groundwater, which is fed by precipitation in the headwaters of the basin."

Line 386-388: As mentioned before, for me, there is no need for an introductory paragraph to each section. The authors may consider to remove these lines.

**Answer:** comment accepted.

**Action taken:** the paragraph has been removed.

Line 401/402: Redundant, see comment before. The authors may consider to remove these lines.

**Answer:** comment accepted.

**Action taken:** these lines have been removed.

Line 482/483: doubling the information that seasonal changes are dominated by radiation (mentioned in line 479/480).

**Answer:** comment accepted.

**Action taken:** we have modified the sentence at line 479/480 as follows: "In analyzing the budget of evaporation at the sub-diurnal scale, the wind speed (aerodynamic contribution) is the main driver of evaporation.", and we have kept the sentence at lines 482/483.

Line 503-505: Consider to merge the two sentences.

**Answer:** We opted to keep these sentences separated to avoid very long and confusing sentences, but we rewrote the text for clarity

**Action taken:** We have modified the second sentence (line 504) as follows: "These correlations express that warm ENSO phases are characterized by higher evaporation rates and lower precipitation, whereas cool phases with lower evaporation and higher precipitation."

Line 587: ice melting occurs when air temperatures are "above" 270K rather than "below"?

**Answer:** comment accepted.

**Action taken:** changed as suggested.

---

## Author Comment (AC2)

Reviewer #2

We thank Dr. Stephanie Kampf for her constructive comments and valuable contribution to our manuscript. Below, we answered the line-by-line comments in blue font and indicated if, how, and where we introduced changes in the revised manuscript in a colored font.

This manuscript is an interesting multi-scale, multi-method evaluation of evaporation and water balance at the Salar del Huasco in Chile. The paper contributes insight into climate drivers of evaporation variability and illustrates how dominant controls on evaporation vary with time scale. The manuscript is well-written, with methods carefully documented.

My suggestion of major revisions is due to concerns about influences on evaporation that appear to be neglected:

1) Salinity reduces evaporation rates, and as far as I can tell this effect is not included in the site-adapted Penman equation. See Mor et al. 2018 WRR.

Answer: Indeed salinity reduces evaporation rates. We demonstrated this for the wet-salt surfaces in the Salar del Huasco (Suarez et al., 2020; Lobos-Roco et al., 2021) and Mor et al., 2018 (Dead Sea) is a good reference as well. In contrast to what Mor et al., 2018 describe for the Dead Sea, in the Salar del Huasco surface runoff is negligible (see also point 2 to our reply of reviewer 1). Due to its shallowness (~15 cm) it is safe to assume that the lake is well-mixed (de la Fuente and Niño, 2010) and the salinity therefore uniform in depth and space.
In our approach, the salinity effect is implicitly included as we fit empirical model constants in our adapted Penman model to the locally measured evaporation fluxes over the saline water surface (see Suárez et al., 2020 and Lobos-Roco et al., 2021). It is important to mention that based on the data available we cannot include the salinity effect based on first-order physical principles. We recall that that is our aim to study the long-term evaporation variability from basic meteorological data provided by ERA5, with which it is not possible to represent the effect of salinity explicitly.

Action taken: we have included a sentence in Appendix A in line 520 to clarify this point: "Note that some physical processes such as the effect of salinity on evaporation, are implicitly included in the site-adapted Penman equation as The empirical coefficients in the model are obtained using evaporation fluxes measured over the saline water surface (Suárez et al, 2020; Lobos-Roco et al., 2021). "

2) Although open water evaporation rates are likely highest, water can also evaporate from areas with salt crusts (see Kampf et al. 2005 JOH, though probably some more recent references are also available). Because the salt crust areas may be large relative to the open water, they likely do have a substantial effect on the basin water balance. An interesting study on salt crust changes over time in Bowen et al. 2017, Geomorphology.

Answer: We fully agree with dr. Kampf. Open water surfaces are not the most extended evaporation pathways in the Atacama desert. Different types of salty crusts (Kampf et al., 2005), which cover larger areas than open water surfaces can contribute significantly to the basin's water balance, although their evaporation rates are considerably lower compared to open waters. The salt crust found in the Salar del Huasco has a particularly low evaporation (< 50 W m$^{-2}$, e.g., see Lobos-Roco et al., 2021, Fig. 3b). In addition, our research focuses on the open water surface (line 88), where we study the multi-scale, temporal changes. Our simple approach to the water balance of the lake does not include other surfaces that can contribute as well to the water balance of the entire basin. In the conclusions section,

we recommend that further research should be carried out to find site adapted Penman coefficients that allow an extension of our method to different surfaces, such as wet salty crusts and wetlands.

**Action taken:** to clarify this point, we have introduced the following sentences in the results and discussion section, line 350:
"Different types of salty crusts (Kampf et al., 2005), which cover larger areas than open water surfaces, can contribute significantly to the basin's water balance, although their evaporation rates are considerably lower compared to open waters. The salt crust found in the Salar del Huasco has particularly low evaporation (< 50 W m$^{-2}$, e.g., see Lobos-Roco et al., 2021, Fig. 3b).".

Please incorporate these effects into the analysis, or explain why they can be neglected.

Other minor suggestions:

line 214: "Evaporation estimates are obtained from the downscaled ERA5 and precipitation" - presumably precipitation data are not used to calculate evaporation. Should this state "precipitation-adjusted evaporation estimates"?

**Answer:** This is indeed confusing. Precipitation data were not used to estimate evaporation. However, these data were used to estimate the long-term water balance in the saline lake. Evaporation estimates were obtained using the site-adapted Penman equation (2.3.2) and the downscaled meteorological data from ERA5. Precipitation was taken from the raw ERA5 data (not downscaled).

**Action taken:** to clarify this, we have introduced the following changes in line 214: "Evaporation estimates are obtained using data from the downscaled ERA5 and the site-adapted Penman equation (section 2.3.2), whereas precipitation data were obtained from the raw ERA5."

line 215: how is the lake depth determined?

**Answer:** The lake depth was measured during the field-experiment, which is described in Suarez et al., 2020 and Lobos-Roco et al., 2021. Consistent depths were also reported by de la Fuente & Meruane 2017 and de la Fuente et al., 2014 in the same saline lake.

**Action taken:** no actions have been taken, as this is well-described in section 2.1 (study area, line 115).

lines 257-258: "we observe that and coefficients". Should the "and" be deleted here, or is another word missing?

**Answer:** comment accepted.

**Action taken:** the word "and" has been removed.

Table 2, 1st row: "addapted"

**Answer:** comment accepted.

**Action taken:** the word "addapted" was changed to "adapted".

Table 2, what is "m" column?

**Answer:** m is the slope of the orthogonal regression.

**Action taken:** we have added this information at the end of the caption of Table 2. "m represents the slope of the orthogonal regression between EC$_{water}$ and estimated through the Penman equation."

Figure 5: Time series are great to see, but I would suggest (1) plotting as lines rather than columns for easier viewing, and (2) paring this with a scatterplot of met station vs ERA5, so the reader can more easily evaluate the performance comparison. Consider also adding precipitation to the time series to visualize how these changes in evaporation correspond with year-to-year and seasonal variability in precipitation. This time series information about precipitation would be a helpful addition to the combined year precipitation data in Fig 6.

**Answer:** We thank Dr. Kampf for her suggestions regarding the visualization of Figure 5. The suggested plots are given below. We do not see a significant difference using lines instead of bars. In fact, we think that the line plot makes it a more difficult to appreciate the difference between the two variables (evaporation from ERA5 & met-stat). We agree that a scatter plot would also be useful for the comparison. However, because of the large number of figures and sub-figures in the manuscript, we opted to simplify this plot by only showing the comparison over the years, which fits better with the message, and include the orthogonal regression coefficients in the text (R$^2$: 0.81 and m , line 273). Finally, we opt not to include precipitation in this plot for two reasons. The first one is that Figure 5 is part of the section "Diurnal cycle perspectives of evaporation", where daily precipitation is not a main driver. The second is that this information is included in Figure 6 with a more robust statistic.

[Figure]

**Action taken:** no actions has been taken.

Figure 9: Similarly, I am curious what these patterns look like as a time series rather than aggregated to monthly means and ranges. The complete time series (or an example series of years) would illustrate how much the lake area changes from year to year & how those area changes relate to precipitation and evaporation.

**Answer:** The Figure below shows the information displayed in Figure 9 as a time series of monthly data. Because of the high variability, it is more difficult to identify the relation between the three components. In addition to this, section where we present Figure 9 is focused on seasonal cycles of evaporation and not interannual ones (the section after). The seasonal cycle of evaporation and its role on the water balance of the saline lake is better represented using monthly aggregated means.

[Figure]

Action taken: no actions have been taken.

Lake water balance, paragraph starting line 369: I am not entirely following the water balance calculations and results. Could you show the water balance graphically?

Answer: Our long-term water balance calculation is clearly explained in lines 213/225 (section 2.3.5). Again to avoid an even larger number of figures and the manuscript's scope we opted not to include a graphic on water balance.

Action taken: no actions have been taken.

Figure 9: b and c are plotting mean monthly values? Related to the comment above about showing full time series - this monthly aggregation illustrates the average role of evaporation in determining lake surface area, but it misses the interannual variability and how precipitation influences area. If the lake surface area lags behind the precipitation because of the slower moving groundwater, then comparing one month's area to the same month's precipitation will not necessarily be helpful. You could try correlations between precipitation and area using the full time series, but instead of comparing same months, lag the lake area month until you find the lag time at which precipitation and lake area are best correlated.

Answer: Please, see the answer to the previous comment in Figure 9 regarding monthly aggregation v/s interannual time series. We acknowledge that there is a lag between precipitation and groundwater recharge, which is possible to see in our data of lake area and precipitation. See the point we make in line 365: "However, analyzing the means (solid lines Fig. 9a), we observe that high precipitation rates do not directly impact the areal changes of the lake, which is reached 4 to 5 months after the rainy season". Here, we stress the point that our water balance of the lake is a first-order approximation, which aim is to show that evaporation plays an essential role in the water balance. Therefore, investigating the role of precipitation in the lake's water balance is beyond the scope of this manuscript.

Action taken: no actions have been taken.

A3: energy balance non-closure coefficient - from Figure A-3, it looks like this is the slope (m) in each scatter relationship? Please connect "m" from the figure to the energy balance non-closure coefficient variable.

Answer: yes, the coefficient is the slope.

**Action taken:** we have introduced the following changes at line 574/575 in Appendix A3: "Since the Penman equation assumes energy balance closure and the E-DATA field data show a significant energy imbalance (Suárez et al., 2020), we introduce an energy balance non-closure coefficient, $c_{EBNC}$, to correct for the observed imbalance, which is the regression slope (m)."

Ice coefficient: on what basis did you choose the number of hours below freezing for ice coefficient values? Did you consider salinity effects on freezing?

**Answer:** the number of freezing hours and their link to an evaporation reduction factor were determined empirically. The idea is that ice produced over longer time periods takes longer to melt. The data that we had available (November 2018) contained days with a maximum of eight hours under freezing conditions. To better underpin this coefficient longer EC time series are needed. Salinity is considered in the analysis since freezing hours are defined by temperatures lower than 270 K and not 273 K as it is for water.

**Action taken:** for clarity, we have introduced the following change at lines 586-587: "For this reason, we use an ice coefficient, $c_{ice}$, which ranges between 0 and 1, depending on the number of freezing hours per day (Table 1). The days are taken from midday to midday to include the night. The idea is that ice produced over longer periods takes longer to melt. We assumed that freezing occurs when the 2 m air temperature is below 270 K, slightly below the freezing temperature of clean water to include the effect of salinity. Based on this criterion, freezing days are distributed over the year as 6% in summer, 21% in fall, 41% in winter, and 31% in spring."

---

## Referee Report (RR1)

The authors have carefully revised the manuscript and well addressed the comments of both reviewers. This significantly improved readability and contributes to clarity of methodological approaches. I have only a few very minor suggestions below, mainly concerning typos. I think that the manuscript can be published in its current form after these minor changes have been made.

Line 127: Change point to comma. "[…] to track the moisture source (Section 2.3.4**,** resulting in precipitation […]"

Line 226: suggested change: "observed and estimated $L_vE$"

Line 232: "introduced **and** coefficients". Either a word is missing or the and should be removed.

Line 241: I think there is something missing in this phrase: "[…] when the radiation decreases yields of $L_vE$."

Figure 4: Suggested change to "average and standard deviation of the diurnal cycle of […]" or "averaged diurnal cycle and standard deviation of […]".

Line 247-248: Does this mean that 30% of the water surface is frozen or 30% of the total water amount?

Line 326: suggested change "**A**mazon basin" (capital letters)

Line 359: suggested change "groun**dw**ater input" (remove space)

Line 360: suggested change "lake **water** balance" instead of mass balance in accordance with change in title of the subsection.

Figure 10: The red line shows not only the evaporation trend, but also that of precipitation. Suggestion to generalize: "The long-term trend is indicated by the red line."

Line 461: suggested change "errors **of** ~7%"

Line 467: suggested change: "are explained **to** 74%"

Line 480: suggest change: "a**n** interannual variability"

Line 507: "[…] are implicitly included in the site-adapted Penman equation as […]" Missing phrase or remove "as".

Line 528: "and ___ are empirical constants"; Symbol missing.

---

## Author Response (AR2)

**Referee #2: Stephanie Kampf, stephanie.kampf@colostate.edu**

The revised manuscript looks excellent, and I have only a few suggestions following on the previous review comments/responses>

We thank Dr. Kampf for her comments. They have improved the quality of our manuscript. Below, we answer her comments in blue font.

1. Regarding the quantity of evaporation from salt crusts, you could calculate what <50 W/m^2 represents in volume relative to volume of evaporation from the open water surface. Although the depth of evaporation from the crusts is small, the surface area of the crust relative to open water looks large. I would be curious to know approximate fractions of evaporation coming from open water vs crust, as that would give you an estimate of the uncertainty in this component of the water balance.

The wet-salt and open-water surfaces have a high seasonal variability, where surfaces continuously change between being open-water or wet-salt (see Fig. 1 in Lobos-Roco et al., 2021, see below). As a result, there is a large uncertainty associated to the contributions of open-water and wet-salt to the total evaporation (expressed as a volume of water) of the basin. We estimate that for November 2018, based on our flux data (4.3 mm/day for open water and 0.5 mm/day for wet-salt) and surface areas estimated from satellite remote sensing observations (1.8 km² for open water and 13.1 km² for wet-salt), the open-water evaporation is 7.8 m³/day against 5.6 m³/day for wet-salt.

In November, the open-water surface is at its minimum extent and the evaporation of the wet-salt reaches to ~40% of the total evaporation volume. We don't have wet-salt evaporation measurements of the summer season (the other extreme) but it is our estimate that the contribution of wet-salt in that season will be below 5% of the total.

[Figure]

**Figure 1.** Shallow saline lake at Salar del Huasco as viewed by the normalized difference water index (NDWI) from Copernicus Sentinel data from 2019 processed by Sentinel Hub. This index combines infrared and visible bands, where dark blue represents water and light green the absence of water. The right-hand image shows the extent of the lake on 18 November 2018, during the field measurements shown in this work.

We have included this analysis in section 3.2.2 by adding the following piece of text in line 337-340:

"Before showing the results, it is interesting to mention the heterogeneous characteristics of open waters and different types of salty crusts (Kampf et al., 2005). These salty crusts cover larger areas than open water surfaces, contributing significantly to the basin's water balance. With respect to the wet/dry salt contribution, the salt crust found in the Salar del Huasco has particularly low evaporation (<50 Wm², Lobos et al., 2021). Despite this low rate, it is still relevant since the open-water/wet-salt surface proportion has high seasonal variability. Based on our flux data (4.3 mm day⁻¹ for open water and 0.5 mm day⁻¹ for wet-salt) and surface areas estimated from satellite remote sensing observations in November (1.8 km² for open water and 13.1 km² for wet-salt), we estimate that open-water evaporation is 7.8 m³ day⁻¹ against 5.6 m³ day⁻¹ for wet-salt. In the rest of the section, we focus on the saline lake water balance as an entity integrating the different contributions. Figure 9 shows ..."

2. Regarding showing time series of monthly data instead of monthly means for Fig 9: although the figure shown in the response is indeed busy as presented, I find it very interesting. The area of the lake is frequently lagged behind precipitation, but not always. Examples in 1987, 1993, 1999-2000, and some others where precipitation is associated with an increase in lake level. Some of these (e.g. 1993) have a precipitation-connected increase in lake level followed by a lagged peak in lake level. My interpretation of this would be that occasionally enough precipitation falls over the lake to increase its level; otherwise, the water source to the lake is groundwater that was recharged elsewhere, then gradually flowed to the lake. You may not wish to present these details in the paper, but pulling out a few example years would be informative to readers wanting to understand the water balance in more detail.

We agree with the observation of Dr. Kampf regarding the time-lag between precipitation and water surface of the lake . Precipitation in the Altiplano region is highly heterogeneous. It normally occurs in the surrounding mountains (Uribe et al., 2015), where rain infiltration in the soil and subsequent upwelling in the lake are months apart. Occasionally, however, as Dr. Kampf rightfully suggests, precipitation occurs close enough to the lake that the much quicker acting surface runoff impacts the lake surface directly. This topic indeed needs further research.

In order to highlight this saline lake water balance effect, we have included a new statement in line 356, including the following sentence:

"In February, the lake surface reaches a first maximum, which might be related to precipitation in the direct proximity of the lake that generates enough surface runoff to enhance the water amount of the lake. However, the highest values of the water-lake surface is reached 4 to 5 months after the rainy season."

3. Regarding the water balance calculation suggestion: Your calculations are well-explained. My confusion here related from bringing a watershed-scale perspective to my reading of this manuscript. I was curious about the size of the area potentially contributing groundwater to this lake and the range of elevation / precipitation differences within that contributing area. For other readers interested in potential groundwater sources, you could provide some context in the water balance description like (lines 231-232) "the additional water source represents groundwater inputs that originated from groundwater recharge in upslope areas of the contributing basin"

We understand the concern of Dr. Kampf related to groundwater. In the revised version of the manuscript, we included a modification in paragraph 2 of section 3.2.2 (see our response to point 2). We highlighted that lake recharge is indirectly related to precipitation if rain occurs in the surrounding mountains and takes 4-5 months to upwells in the lake.

**Referee #1: Claudia Voigt, c.voigt@uni-koeln.de**

The authors have done a good job and carefully revised the manuscript. I suggest to accept the manuscript as is after very minor technical corrections.

The authors have carefully revised the manuscript and well addressed the comments of both reviewers. This significantly improved readability and contributes to clarity of methodological approaches. I have only a few very minor suggestions below, mainly concerning typos. I think that the manuscript can be published in its current form after these minor changes have been made.

We thank Dr. Voigt for her detailed review that has led to improve the quality of our manuscript. Below, we have answered in blue font her comments.

Line 127: Change point to comma. "[…] to track the moisture source (Section 2.3.4), resulting in precipitation […]"

Changed as suggested

Line 226: suggested change: "observed and estimated LvE"

Changed as suggested

Line 232: "introduced and coefficients". Either a word is missing or the and should be removed.

The word "and" has been removed

Line 241: I think there is something missing in this phrase: "[…] when the radiation decreases yields of LvE."

The sentences has been modified by "The radiative energy control is more clearly observed from 14:00-15:00 LT when radiation decreases the $L_vE$ yields"

Figure 4: Suggested change to "average and standard deviation of the diurnal cycle of […]" or "averaged diurnal cycle and standard deviation of […]".

The sentence has been modified by "Daily average and standard deviation of $L_vE$ observed by the $EC_{water}$ and calculated by $P_{SDH}$ equation during the E-DATA period".

Line 247-248: Does this mean that 30% of the water surface is frozen or 30% of the total water amount?

The ice coefficient is applied to each single day, therefore the total water amount. The idea behind it is that the longer the freezing hours, the more time takes to melt the ice before evaporating water.

Line 326: suggested change "Amazon basin" (capital letters)

Changed as suggested

Line 359: suggested change "groundwater input" (remove space)

Changed as suggested

Line 360: suggested change "lake water balance" instead of mass balance in accordance with change in title of the subsection.

Changed as suggested

Figure 10: The red line shows not only the evaporation trend, but also that of precipitation. Suggestion to generalize: "The long-term trend is indicated by the red line."

Changed as suggested

Line 461: suggested change "errors of ~7%"

Changed as suggested

Line 467: suggested change: "are explained to 74%"

Changed as suggested

Line 480: suggest change: "an interannual variability"

Changed as suggested

Line 507: "[…] are implicitly included in the site-adapted Penman equation as […]" Missing phrase or remove "as".

We have removed the as and added the missing period.

Line 528: "and ____ are empirical constants"; Symbol missing

We have added the missing empirical constants as " $c_1 = 0.05$ and $c_2 = 2.45$ are empirical constants".

---

## Author Response (AR3)

**Comments to the author:**
Thanks for your revisions. Just a few technical points:

We thank the editor for indicating these technical points. Below we answer them in blue font.

Lake's area should be Lake area, I think

Changed as suggested

Why Pp for Precipitation when you use single letters for the other terms (e.g., E)? Perhaps writing the full word on the axes might be even better

Thanks for the advice. We use "Pp" for precipitation and not a single letter like "P" to not confuse the "P" of Penman (e.g., $P_{SDH}$, $P_{stdr}$) in section 3.1 (Figure 4). We prefer to not use the full word to avoid extended axis labels and to be consistent with other abbreviations like E, ONI, PDO, etc. To avoid confusion, we have added next to the words precipitation and evaporation (Pp) and (E), respectively, in the caption of figures 6, 9, 10, 12, and 13.

A1: r_a, add Resistance on axis? local with capital L

I think that the editor refers to Fig A2. Changed as suggested.

A2/A3: is it clear that m is slope?

It is explained in line 571. In addition, we have added " (m) corresponds to the regression slope" in the caption of Figure A3.

A4: I would say degree hours have the unit h, not? Ta and similar elsewhere (e.g. Rn): doublecheck the use of subscripts

Yes, it was changed as suggested and all subscripts have been revised.